# Variation in the fruit development gene *POINTED TIP* regulates protuberance of tomato fruit tip

Jianwen Song [1], Lele Shang [1], Changxing Li[1], Wenqian Wang[1], Xin Wang [1,2], Chunli Zhang [1], Guo Ai[1], Jie Ye [1], Changxian Yang[1], Hanxia Li[1], Zonglie Hong [3], Robert M. Larkin[1], Zhibiao Ye [1] & Junhong Zhang [1,2] ✉

The domestication of tomato has led to striking variations in fruit morphology. Here, we show a genome-wide association study (GWAS) to understand the development of the fruit tip and describe a *POINTED TIP* (*PT*) gene that encodes a C2H2-type zinc finger transcription factor. A single nucleotide polymorphism is found to change a histidine (H) to an arginine (R) in the C2H2 domain of PT and the two alleles are referred to as $PT^H$ and $PT^R$. Knocking out $PT^H$ leads to development of pointed tip fruit. $PT^H$ functions to suppress pointed tip formation by downregulating the transcription of *FRUTFULL 2* (*FUL2*), which alters the auxin transport. Our evolutionary analysis and previous studies by others suggest that the $PT^R$ allele likely hitch-hiked along with other selected loci during the domestication process. This study uncovers variation in *PT* and molecular mechanism underlying fruit tip development in tomato.

Most fruits are specialized organs that develop from ovaries after successful pollination and fertilization. Fruits protect the developing embryos and promote the dispersal of mature seeds[1]. Fruits contribute to human health by providing a variety of compounds that are nutritionally valuable, such as carbohydrates, vitamins, and fiber. Due to its agronomic value and the intense research on its agronomic traits, tomato (*Solanum lycopersicum*) has become a model species for studying fleshy fruit[2,3]. It is generally believed that the tomato originated in the Andes region of South America and was initially domesticated in Central America[4]. Recently, genome sequences and agronomic traits analyses of tomato accessions originating from Latin America have suggested that the history of tomato domestication contains a complex and dynamic intermediate stage[5]. The wild relatives of tomatoes produce a small round fruit. However, during the domestication and improvement of tomato, remarkable morphological fruit diversity, especially in fruit shape and size, was selected and preserved in cultivated tomatoes[4]. Based on their morphologies, tomato fruits can be divided into round, heart, oxheart, long, flat, rectangular, ellipsoid, and obovoid[6].

Among the genetic loci underlying fruit morphologies in tomato, several genes have recently been cloned. Locule number, which is positively correlated with the flat degree of fruit shape, has been found to be controlled by the *locule number* (*lc*) and *fasciated* (*fas*) loci. Mutations in these loci have been attributed to two single-nucleotide polymorphisms (SNPs) that are located ~1.2 kb downstream of the *WUSCHEL* gene and a 294-kb inversion that is located 1 kb upstream of the *CLAVATA* (*CLV3*) gene, respectively[7–9]. *SUN* encodes an IQ67 domain-containing protein that promotes fruit elongation. A gene duplication event mediated by a *Rider* retrotransposon leads to increased *SUN* expression in the *sun* mutant[10]. *OVATE* encodes an OVATE family protein (OFP) that attenuates fruit length. A mutation that introduces a premature stop codon in *OVATE* induces a transition from the development of round to pear-shaped fruit[11]. However, some tomato varieties still develop round fruit in the *OVATE* mutant background. Thus, additional genes may

[1]Key Laboratory of Horticultural Plant Biology (MOE) and National Center for Vegetable Improvement (Central China), Huazhong Agricultural University, Wuhan 430070, China. [2]Hubei Hongshan Laboratory, Wuhan 430070, China. [3]Department of Plant Sciences, University of Idaho, Moscow, ID 83844, USA. ✉e-mail: zhangjunhng@mail.hzau.edu.cn

also contribute to the development of particular fruit shapes[6]. Recently, a suppressor of ovate (*OFP20*), encoding another OFP family member, has been cloned and found to regulate fruit shape by interacting with a TONNEAU1 recruiting motif (TRM5) protein and affecting the localization of protein complexes[12].

In heart-shaped fruits, the proximal end is larger than the distal end, the prominent shoulders are reduced relative to the oxheart fruit, and the distal end has a distinctly pointed tip[6]. About half of the heart-shaped tomato varieties harbor a mutant allele of *OVATE*. In contrast, the *ovate* mutant develops pear-shaped fruit[6]. Thus, the development of heart-shaped tomato fruit is controlled by other loci. The most characteristic feature of heart-shaped tomatoes is the pointed tip at the distal end, which is different from the phenotypes of other fruit-shaped mutants. Rough scars at the distal end of fruit severely affect the quality of the tomato. Selection of accessions with pointed tip genotype is used in breeding to eliminate the rough blossom-end scarring in large-fruited, fresh-market tomatoes[13]. Previous studies have identified seven phenotypically similar mutants (i.e., mutants that develop pointed blossom-end fruit), including *beaked* (*bk*), *bk-2*, *nipple-tip* (*n*), *n-2*, *n-3*, *n-4*, and *persistent style* (*pst*)[13]. *bk*, *n*, and *pst* have been mapped to chromosomes 2, 5, and 7, respectively[14–16]. However, the *bk*, *n*, and *pst* alleles have not been cloned.

Using the reverse genetic approach, a few genes have been implicated in the development of fruit with pointed tips. *SlGGB1* encodes a γ-subunit of type B heterotrimeric G proteins. Silencing of *SlGGB1* leads to the production of fruit with pointy tips, which is possibly mediated by alterations in auxin signaling[17]. Overexpression of *SlFUL2*, a MADS transcription factor, produces an abnormally persistent style, and the abnormal abscission of the pistil leads to ovary cell development after the pollination and fertilization stage and results in the formation of pointed tip fruits[18]. Additionally, auxin also directly influences the development of fruit with pointed tips in tomato. RNA interference experiments that downregulate the expression of auxin response factor 7 (ARF7) led to the production of fruit with heart-like shapes, probably by strongly influencing auxin signaling[19]. The fruit produced by *DefH9-iaaM* transgenic lines that accumulate high levels of auxin have the "pickelhauben" phenotype with a pointed tip at the distal end[20]. Even though auxin has been implicated in the development of fruit with pointed tips, the details of the mechanism are largely unknown.

C2H2-type zinc finger proteins are widely distributed in plants. Classical C2H2-type zinc finger proteins contain a specific conserved domain, defined as $CX_{2-4}CX_{12}HX_{2-6}H$[21]. They often serve as transcription factors and play crucial roles in the development and stress responses[22,23]. The expression of the Arabidopsis C2H2 transcription factor *Zat12* responds broadly to biotic and abiotic stress. The expression of *Zat12* is specifically enhanced when plants are exposed to intense light and when plants experience oxidative stress[24]. In rice, a natural variation in the promoter of *bsr-d1* reduces its expression, inhibits the degradation of $H_2O_2$, and enhances broad-spectrum blast resistance[25]. In *Arabidopsis*, *KNUCKLES* (*KNU*) might function as a transcriptional repressor of cellular proliferation that regulates floral determinacy and basal pattern elements because the *knu* mutant develops flowers and siliques with abnormal morphologies[26]. Previous studies have predicted 112 C2H2-type transcription factors in tomato[27], and they have been implicated in multiple processes, such as *ZF3* in salt tolerance[28], *Hair* in trichome formation[29], *SE3.1* in flower development[30], *OBV* in leaf veins development[31]. However, the natural variation of C2H2 zinc finger proteins and their influence on fruit shape in natural populations remains largely unknown.

Here, we perform GWAS to identify genes that may contribute to the development of pointed tips in tomato fruit, taking advantage of the natural variation in a collection of germplasms containing 311 accessions of tomato. A locus, designated as *POINTED TIP* (*PT*), is

discovered and found to encode a C2H2 transcription factor. This locus has also been identified recently to underlie the "obscure vein" (*OBV*)[31]. A nucleotide substitution from A to G at the lead SNP generates two alleles of *PT* (*PT^H* and *PT^R*) that control the fruit blossom-end shape from a non-pointed tip to a pointed tip shape. Analysis of results from the overexpression and knockout of *PT* reveals that *PT^R* and *PT^H* are responsible for the formation of fruit with pointed tips or non-pointed tip shapes, respectively. Therefore, our research identifies a variation in the conserved C2H2 domains of PT that affects downstream gene expression and regulates the determination of fruit shape in tomato.

## Results

### Identification of a major locus regulating the pointed tip fruit trait

To understand the genetic basis for the morphology of the distal end of tomato fruit, we collected a diversity panel of 311 tomato accessions, which included one accession of *Solanum galapagense* (wild), one *Solanum neorickii* (wild), two *Solanum cheesmaniae* (wild), 51 *Solanum pimpinellifolium* (PIM, wild), 105 *S. lycopersicum* var. *cerasiforme* (CER), and 151 *S. lycopersicum* (BIG) (Fig. 1a and Supplementary Data 1). Based on the angle (θ) of the distal end of the fruit, we classified the entire collection into two groups, fruit with pointed tips (0 ° <θ < 180°) and non-pointed tips fruit (Fig. 1c, d and Supplementary Data 1). Although the wild tomato accessions did not produce fruit with pointed tips, as many as 21.85% of the BIG accessions produced fruit with pointed tips (Fig. 1b).

Using 4,155,820 SNPs (MAFs > 0.05, the number of varieties with minor alleles ≥ 16) covering the entire tomato genome, we performed GWAS on the fruit morphology of the distal end phenotype (non-pointed tip: 0, pointed tip: 1; Supplementary Data 1), and identified a locus containing the lead SNP (ch05_64012700, $P = 7.57 \times 10^{-40}$) on chromosome 5 that was significantly associated with the development of fruit with pointed tips (Fig. 1e, f and Supplementary Data 2). The linkage disequilibrium (LD) block containing the lead SNP, ch05_64012700 ($P = 7.57 \times 10^{-40}$), harbored a single gene, which encoded a C2H2-type zinc finger protein (Solyc05g054030; Fig. 1g–i). Interestingly, the lead SNP is located in an exon of Solyc05g054030 (Fig. 1g–i). The guanine (G) allele was associated with the development of fruit with pointed tips. The adenine (A) allele was associated with non-pointed tip fruit (Fig. 2b and Supplementary Data 1). Thus, we hypothesized that Solyc05g054030 was the best candidate gene and provisionally named it *POINTED TIP* (*PT*).

To test whether we correctly identified the *PT* locus, we crossed an accession that produces non-pointed tip fruit (TS-9) and accession that produces fruit with pointed tips (LA4053) and constructed an F2 population (Fig. 1j, k). The F1 generation from this cross-produced non-pointed tip fruit. The pointed tip phenotype is segregated in the F2 generation. To identify the gene responsible for the pointed tip trait, we performed bulk segregant analysis (BSA) with individuals from the F2 population and sequenced pools of tissue from individuals that produced non-pointed tip fruit and individuals that produced fruit with pointed tips. Each pool contained 73 individuals. The sequencing depth of each pool was 70 × genome equivalents. Based on the differences in the allele frequencies, the loci of the pointed tip fruit were mapped to chromosomes 2, 3, and 5 (Fig. 1l). The locus on chromosome 5 (Chr5: 63.05-65.05 Mb; Fig. 1m) overlapped the LA4053 allele (G allele) of *PT*, which leads to the development of fruit with pointed tips and is consistent with the results from the GWAS. Therefore, based on the GWAS and the BSA analysis, we concluded that Solyc05g054030 is the causal candidate gene for the development of fruit with pointed tips and that the lead SNP G/A (ch05_64012700) located in Solyc05g054030 is probably responsible for the variation in the distal end morphology of the fruit.

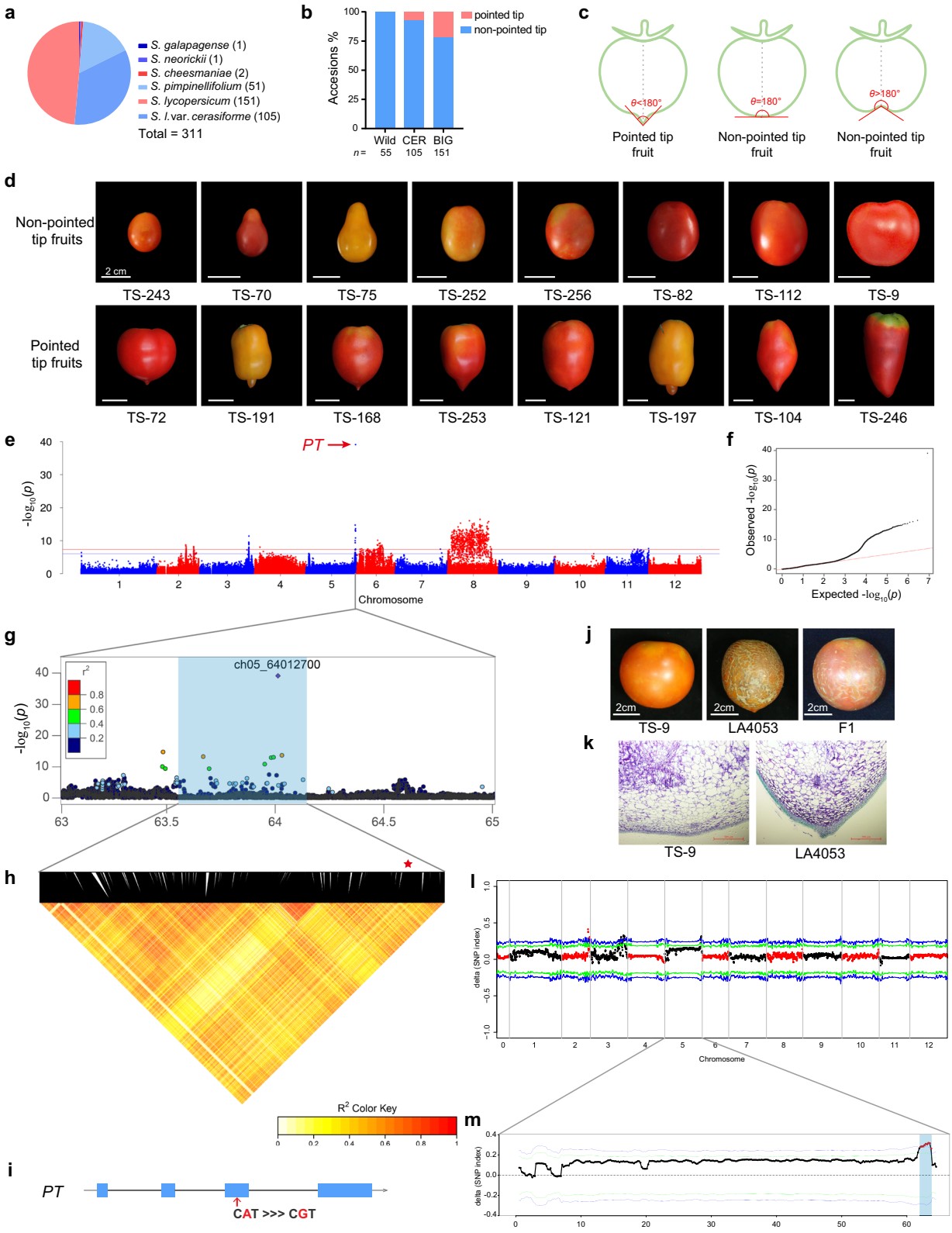

## A single-nucleotide polymorphism is predicted to alter the number of C2H2 domains in the PT protein

To study the variations in *PT*, we used Sanger sequencing to compare the sequences of Solyc05g054030 in accessions that produce fruit with a pointed tip and non-pointed tip. We found that the SNP (1593 $^{G/A}$) with the highest value (ch05_64012700, $P = 7.57 \times 10^{-40}$) is located in the third exon of Solyc05g054030 and changes a histidine residue (H-

135) to an arginine (R-135), thus generating two alleles of *PT* ($PT^H$ and $PT^R$; Figs. 1i, 2c and Supplementary Data 3). H-135 is highly conserved in the C2H2 domain. Moreover, substituting H-135 for R-135 changes the number of C2H2 domains in the PT protein. Specifically, there are two C2H2 domains in $PT^R$ and three C2H2 domains in $PT^H$ (Fig. 2a, c and Supplementary Data 3). Phylogenetic analysis of C2H2 transcription factor proteins from plants shows that the tomato PT protein is located

**Fig. 1 | Identification of major loci for the pointed tip trait using GWAS and BSA.**
**a** Composition of the 311 tomato accessions. **b** Percentages of fruit with pointed and non-pointed tips in wild, CER, and BIG accessions. **c** Schematic diagram of pointed tip and non-pointed tip fruit. $\theta$ represents the angle of the distal end of the fruit. **d** Representative images of fruit with pointed and non-pointed tips. Bars, 2 cm. **e** Manhattan plot from GWAS on the morphology of the distal end of the fruit. The blue and red dashed lines indicate the Bonferroni-adjusted significant and suggestive thresholds ($P = 2.41 \times 10^{-7}$ and $1.20 \times 10^{-8}$). The lead SNP (ch05_64012700, $P = 7.57 \times 10^{-40}$) in the *PT* gene is indicated with a red arrow. **f** Quantile-quantile plot of expected versus observed *P* values for GWAS. **g** Detail plot surrounding the lead SNP (purple rhombus) on chromosome 5. The color of each dot corresponds to the $r^2$ value indicated on the color scale. **h** Representation of the pairwise $r^2$ values among all polymorphic sites in the genomic region (63.48–64.03 Mb) from **g**. The haploblocks are represented by inverted triangles.

The darkness of the color of each box indicates an $r^2$ value indicated in the color scale. The red star indicates the lead SNP containing haploblock. **i** Gene structure in the haploblock containing the lead SNP. Blue boxes, black lines, and arrows represent exons, introns, and gene orientation, respectively. The lead SNP is indicated with red under the blue box. **j** Fruit phenotypes of TS-9, LA4053, and their F1 progeny. **k** Cells from the distal end of the ripening fruit from TS-9 and LA4053. Three independent experiments were performed. **l** Distribution of the ΔSNP-index between the fruit with pointed and non-pointed tips. The statistical confidence interval under the null hypothesis of no QTLs is indicated with green ($P < 0.05$) and blue ($P < 0.01$) lines. **m** Details on loci that produce fruit with pointed tips on chromosome 5. Significance windows for the ΔSNP-index above the confidence intervals are indicated with red points and blue shading. Source data underlying Fig. 1b are provided as a Source Data file.

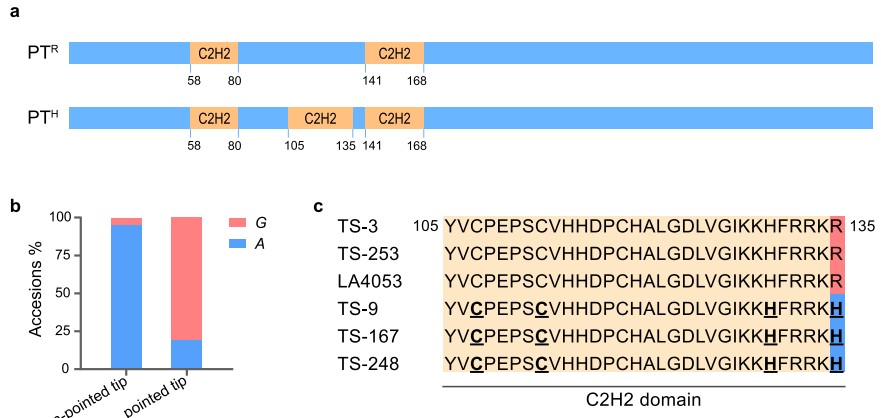

**Fig. 2 | The G/A polymorphism in *PT* leads to the change in the C2H2 domain number. a** Schematic diagram of $PT^R$ and $PT^H$ alleles. The protein structures were predicted using the SMART database (http://smart.embl-heidelberg.de/). Orange boxes indicate C2H2 domains. **b** Percentages of the *G* and *A* alleles in *PT* genes from the accessions that produce fruit with pointed tips and non-pointed tips. **c** Amino acid sequence alignment from residue 105 to 135. TS-3, TS-253, and LA4053

produced fruit with pointed tips. TS-9, TS-167, and TS-248 produced non-pointed tip fruit. The underlined amino acids marked in bold are the conserved C and H residues of the C2H2 domain. The amino acid variation that substitutes R and H residues is highlighted with pink and blue backgrounds, respectively. Source data underlying Fig. 2b are provided as a Source Data file.

in an independent branch, implying that *PT* may have distinctive and unique functions in tomato (Supplementary Fig. 1). We then tested the subcellular localization of PT$^R$ and PT$^H$ in protoplasts that we prepared from the leaves of *Nicotiana benthamiana*. We found that the variation in the C2H2 domain did not affect the subcellular localization of PT$^{R/H}$ and that both of them accumulated in the nucleus (Supplementary Fig. 2). Transcription factors can bind specific DNA sequences in gene promoters. The changes in the number of C2H2 domains may affect the affinity of the PT protein for particular DNA sequences and, therefore, alter the ability of PT to regulate the transcription of particular genes. We also sequenced the promoter region of Solyc05g054030 and found no variations in the pointed tip accession LA4053 relative to the non-pointed tip accession TS-9 (Supplementary Fig. 3).

Additionally, we analyzed the transcript levels of *PT* in pointed tip fruit accessions and non-pointed tip fruit accessions. We found that *PT* was transcribed at similar levels in the pointed tip and non-pointed tip fruit accessions. In both groups of accessions, *PT* was constitutively expressed in all organs tested, including the root, stem, leaf, flower, style, ovary, and fruit (Supplementary Fig. 4). In fact, *PT* was most highly expressed in the leaf, flower, style, and early immature fruit (3–7 DPA) of both pointed tip and non-pointed tip fruit accessions (Supplementary Fig. 4). Combining the variations and transcript levels analysis, we concluded that the lead SNP in *PT* alters the number of C2H2 domains in the PT protein and that this lead SNP is probably responsible for the formation of fruit with pointed tips.

## Loss-of-function mutation in *PT* leads to the formation of pointed tips in fruit

To independently test whether *PT* regulates the morphology at the distal end of tomato fruit, we studied the impact of the $PT^{R/H}$ allele in an F2 population. Using cleaved amplified polymorphic sequence (CAPS) markers derived from our BSA, we screened the individual plants that contained only the $PT^H$ or only the $PT^R$ locus on chromosome 5 and constructed the F2:3 population (Supplementary Table 1). Compared to the individual plants harboring the $PT^H$ allele in the F2:3 population, the individual plants harboring the $PT^R$ allele produced fruit with pointed tips (Fig. 3a and Supplementary Figs. 5a, 6a). Furthermore, we transformed *35 S:PT$^H$* into the pointed tip fruit accession (TS-3) that is homozygous for the $PT^R$ allele and non-pointed tip fruit accession (TS-9, $PT^H+/+$), respectively. We observed that the $PT^H$-overexpressing lines (PT$^H$-OE-2, PT$^H$-OE-3, and PT$^H$-OE-5) produced fruit with non-pointed tips relative to its wild-type TS-3 (Fig. 3d and Supplementary Figs. 5b, 6c). In fact, the overexpression of $PT^H$ (PT$^H$-OE-5, PT$^H$-OE-7, and PT$^H$-OE-8) in TS-9 ($PT^H+/+$) and overexpression of $PT^H$ (PT$^H$-OE-2, PT$^H$-OE-3, and PT$^H$-OE-5) in TS-3 ($PT^R+/+$) led to the development of oval-shaped fruit and necrosis at the distal end of the fruit (Fig. 3b–d and Supplementary Figs. 5b, c, 6b, c). These data suggested that there is an additive effect of $PT^H$ on inducing increases in the transverse diameter and inhibiting growth at the distal end of the fruit. Overall, these data indicate that the single-nucleotide polymorphism in the $PT^{R/H}$ alleles regulates the morphology at the distal end of tomato fruit.

To independently evaluate the contribution of *PT* to the development of pointed tips in fruit, we selected accessions that produce

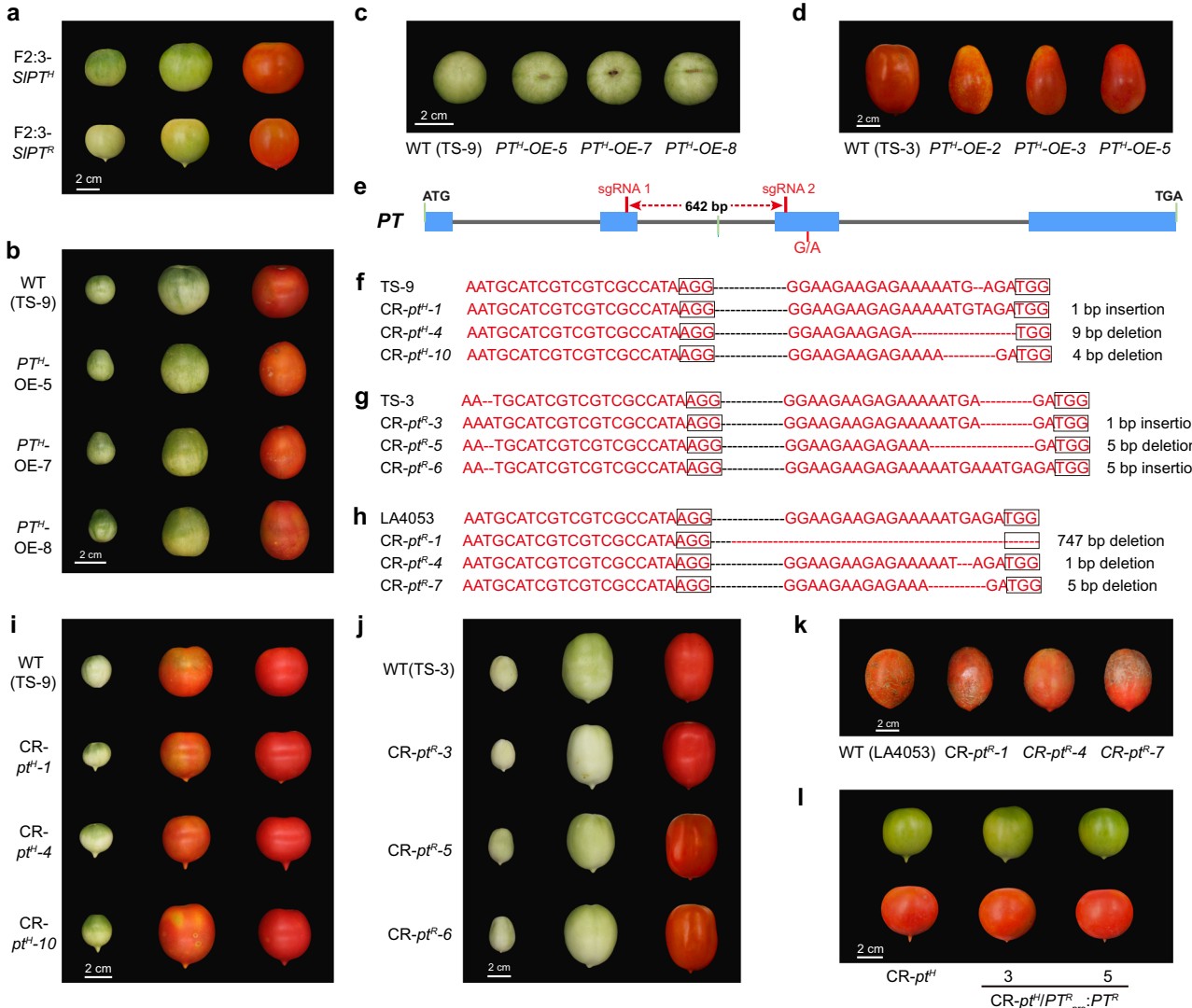

**Fig. 3 | Mutation of *PT* led to the formation of fruit with pointed tips. a** Fruit phenotypes of plants harboring *PT^R* and *PT^H* alleles from the F2:3 population. **b** Fruit phenotypes of *PT^H* overexpressing lines and wild-type (TS-9, *PT^H* allele). **c** Morphology of the distal end of fruit from the *PT^H* overexpressing lines and wild-type (TS-9). **d** Fruit phenotypes of *PT^H* overexpressing lines and wild-type (TS-3, *PT^R* allele). **e** Schematic illustration of two sgRNA target sites in *PT*. The red lines indicate sgRNAs. **f–h** Genomic DNA sequences of *PT^H* mutants (**f**: CR-*pt^H*-1, CR-*pt^H*-4, and CR-*pt^H*-10), *PT^R* mutants (**g**: CR-*pt^R*-3, CR-*pt^R*-5, and CR-*pt^R*-6; **h**: CR-*pt^R*-1, CR-*pt^R*-4,

and CR-*pt^R*-7) and the pertinent wild-type sequences. The black dotted lines indicate the base sequences between sgRNAs. TS-9 harbors the *PT^H* allele and produces non-pointed tip fruit. TS-3 and LA4053 harbor the *PT^R* allele and produce fruit with pointed tips. **i–k** Phenotype of fruit from the CR-*pt^H* mutants (**i**: CR-*pt^H*-1, CR-*pt^H*-4, and CR-*pt^H*-10), CR-*pt^R* mutants (**j**: CR-*pt^R*-3, CR-*pt^R*-5, and CR-*pt^R*-6; **k**: CR-*pt^R*-1, CR-*pt^R*-4, and CR-*pt^R*-7) and the pertinent wild-type. **l** Phenotype of fruit from the CR-*pt^H*/*PT^R*_pro_:*PT^R* lines and the pertinent control CR-*pt^H*.

fruit with pointed tips and accessions that produce non-pointed tip fruit and used CRISPR/Cas9-mediated genome editing technology to create loss-of-function mutations in the endogenous *PT* gene (Fig. 3e). We selected three independent CR-*pt* mutants from each genetic background and sequenced the target region of the genomic DNA of CR-*pt^H*-1, CR-*pt^H*-4, and CR-*pt^H*-10 from TS-9 (non-pointed tip); CR-*pt^R*-3, CR-*pt^R*-5, and CR-*pt^R*-6 from TS-3 (pointed tip); and CR-*pt^R*-1, CR-*pt^R*-4, and CR-*pt^R*-7 from LA4053 (pointed tip). All selected lines contained either an insertion or a deletion in the target region of the *PT* gene that changed the reading frame or deleted amino acids in the conserved spacer between zinc fingers and were thus, knockout mutants (Fig. 3f–h and Supplementary Data 4). We observed that both the CR-*pt^H* and CR-*pt^R* mutants developed pointed tips at the distal end of the fruit (Fig. 3i–k and Supplementary Figs. 5d–f, 6d–f). These data indicate that the *PT* gene function is required for the development of non-pointed tip fruit. Furthermore, we transformed the *PT^R*_pro_:*PT^R* into the CR-*pt^H* plants and found that the CR-*pt^H*/*PT^R*_pro_:*PT^R* still produced fruit

with pointed tips relative to CR-*pt^H* plants (Fig. 3l and Supplementary Figs. 6g, 7g). Taken together, these results indicate that loss-of-function mutations in the *PT* gene lead to the formation of fruit with pointed tips and that the *PT^H* allele exerts its function to suppress the development of pointed tips and leads to the formation of a non-pointed tip fruit in tomato.

## The formation of pointed tips is initiated after anthesis and before style abscission

To pinpoint the time of pointed tip initiation, we dissected developing flowers and fruits at different developmental stages from CR-*pt^H* line and its wild-type control TS-9. The developmental process was divided into stages of flower bud, anthesis, style senescence, and early stage of fruit development (Fig. 4a). Compared with wild-type TS-9, the ovary of CR-*pt^H* displayed no visible differences before and at anthesis. Subsequently, with the senescence of the style (wilted from the tip to the base), the distal end of the ovary from CR-*pt^H* gradually grew to

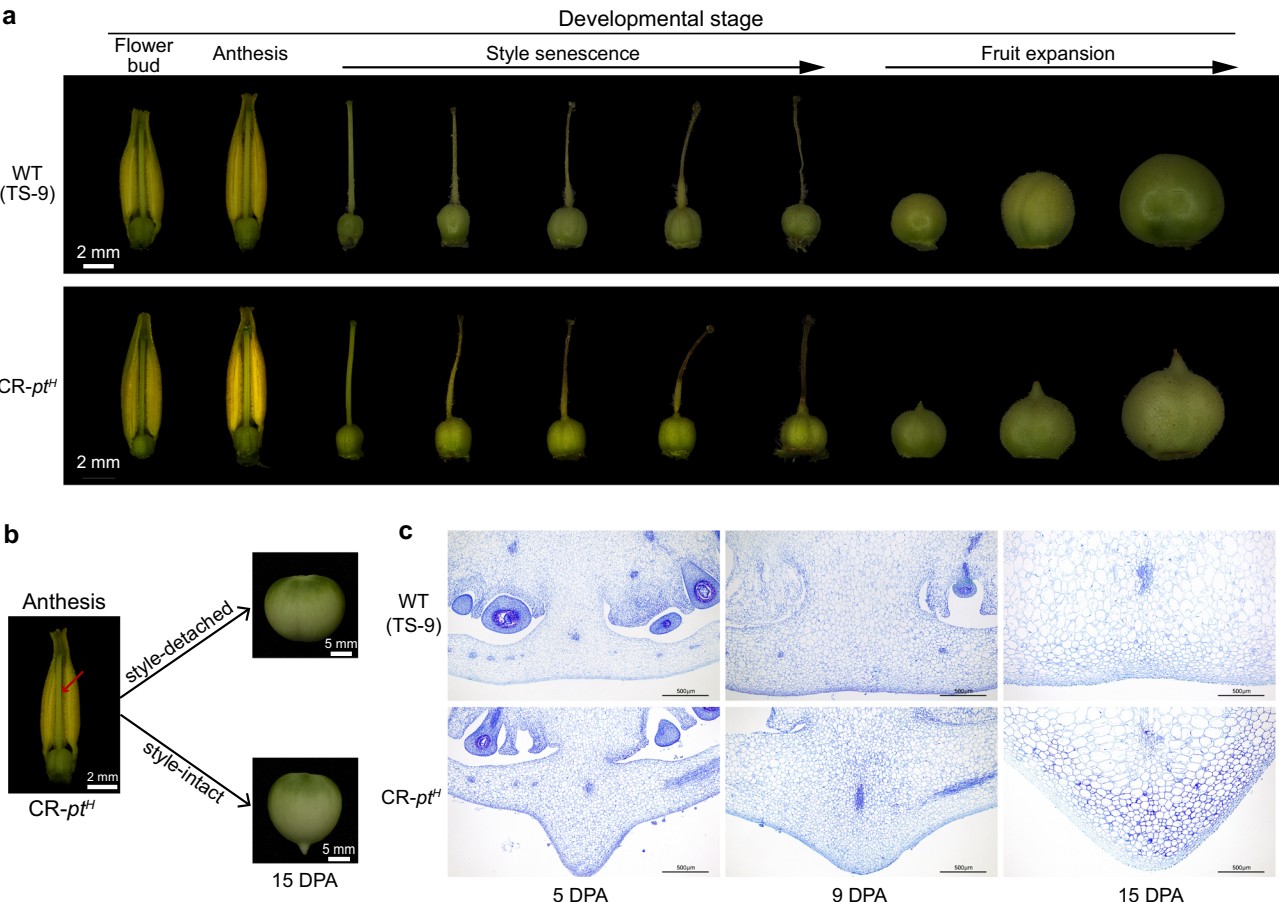

**Fig. 4 | Morphology of the distal end of fruit at different developmental stages.**
**a** Morphology of the distal end of fruit at the stages of flower bud, anthesis, style senescence, and fruit expansion from CR-*pt^H* lines and wild-type (TS-9).
**b** Morphology of the distal end of fruit from CR-*pt^H* lines after detaching or preserving the style. The style was detached after pollination and fertilization.
**c** Longitudinal sections of the distal end of fruit from CR-*pt^H* line and wild-type (TS-9) at different fruit developmental stages. DPA day post anthesis. Three independent experiments were performed.

form the fruit pointed tips, whereas the distal end of the TS-9 ovary remained unchanged (Fig. 4a), indicating that the fruit pointed tip of CR-*pt^H* is initiated after pollination and fertilization and before the style abscission. Furthermore, we detached the style from the pollinated and fertilized flowers in the CR-*pt^H* line to explore the relationship between the presence of a style and the initiation of pointed tips. We found that the style-detached CR-*pt^H* line produced non-pointed tip fruit relative to the style-intact CR-*pt^H* fruit, which formed pointed tips. This experiment suggested that style senescence is required for the formation of the fruit with pointed tips (Fig. 4b). To gain insight into the development of fruit with pointed tips, we performed an anatomical analysis of ovaries at different developmental stages using paraffin-embedded sections. We discovered no apparent abscission zone in the pointed tips, and the development of pointed tip cells originated from the division and expansion of the cells at the distal end of the CR-*pt^H* ovary (Fig. 4c). Based on this, we concluded that the formation of fruit pointed tips of CR-*pt^H* was initiated after anthesis and before the style abscission, and this process was regulated by the style senescence and ovary development.

### *PT^H* suppresses the development of pointed tips by inhibiting the expression of *FUL2*

To gain insight into the mechanism of *PT* in controlling the development of fruit with pointed tips, we conducted RNA sequencing (RNA-seq) in the distal end of the fruit from CR-*pt^H* and wild-type TS-9. The transcript levels of 2938 genes were increased or decreased more than twofold in CR-*pt^H* lines relative to the wild-type TS-9 (Supplementary

Data 5). Gene Ontology (GO) enrichment analysis revealed that *PT* might regulate genes associated with the terms "DNA-binding transcription factor activity", "microtubule binding", and "auxin efflux transmembrane transporter activity" that are involved in "regulation of transcription", "mitotic cell cycle phase transition", "auxin efflux", "auxin polar transport", "auxin homeostasis", and response to "light stimulus" (Supplementary Fig. 7).

The overexpression of *FUL2* also leads to the formation of fruit with pointed tips (Supplementary Fig. 8)[18]. As expected, the transcript level of *FUL2* was significantly increased in CR-*pt^H* relative to wild-type TS-9 in the RNA-seq analysis (Supplementary Data 5). Therefore, we independently quantified the transcript levels of *FUL2* in transgenic lines (TS-9) and found that the transcript level of *FUL2* was upregulated in the CR-*pt^H* mutants and downregulated in the *PT^H*-overexpressing lines (Fig. 5a). These data indicate that *PT^H* may negatively regulate the transcription of *FUL2*. To test this hypothesis, we performed a yeast one-hybrid assay using the promoter (1504 bp upstream of the transcription start site) of *FUL2* and PT proteins. Yeast cells co-transformed with pAbAi-FUL2 and pGADT7-PT^H were able to grow on the SD/-Ura-Leu selective medium containing 50 ng mL$^{-1}$ aureobasidin A (AbA). However, yeast cells co-expressing pAbAi-FUL2 and pGADT7-PT^R could not grow on the same highly stringent selective medium (50 ng mL$^{-1}$ AbA) but could grow weakly on a less stringent selective medium, SD/-Ura-Leu containing 40 ng mL$^{-1}$ AbA (Fig. 5b, c). The growth rates of yeast cells expressing PT^H and the positive control were similar (Fig. 5c). These data provide evidence that PT^H can bind the promoter of *FUL2* with higher affinity than PT^R.

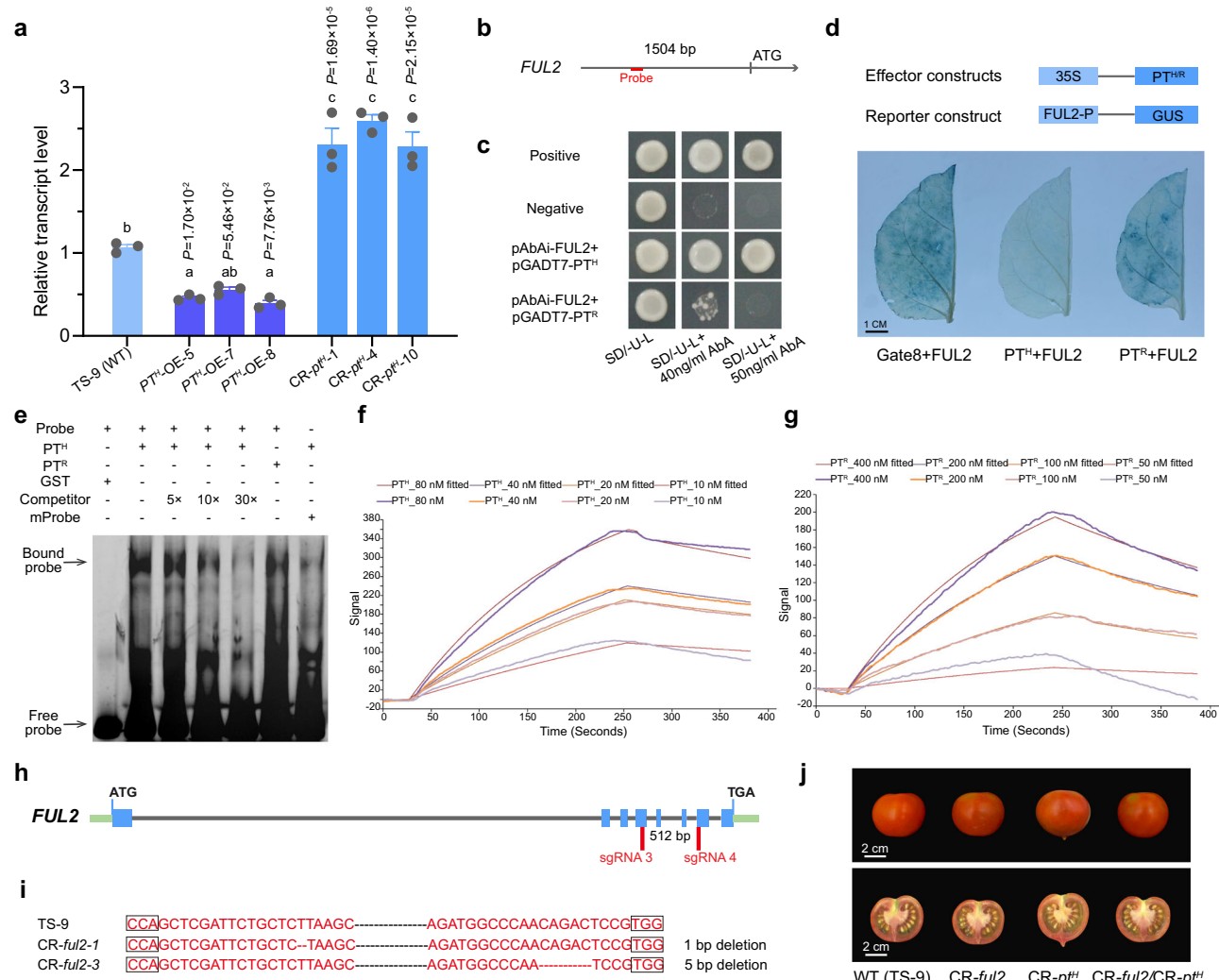

**Fig. 5 | Influence of PT^H-regulated *FUL2* expression on the development of fruit with pointed tips. a** Relative transcript levels of *FUL2* in the distal end of fruit from *PT^H*-overexpressing lines, *PT^H* mutants, and the wild-type control (TS-9, *PT^H* allele). The fruit were harvested at 14 DPA. Error bars indicate mean ± SE. *n* = 3 biological replicates. Statistically significant differences were determined using a one-way ANOVA with Tukey's post hoc test. Different letters indicate statistically significant differences (*P* < 0.05). **b** Schematic diagram of the *FUL2* promoter region. The red line indicates the probe used in **e**. **c** PT^H binding of the *FUL2* promoter in yeast. Positive control: p53-AbAi + pGAD-Rec2-53. Negative control: pAbAi-FUL2 + pGADT7. The transformants were grown on a selective medium (SD/-Ura-Leu) containing 40 or 50 ng mL⁻¹ aureobasidin A (AbA). **d** Co-expression of the PT^H/R and the *FUL2* promoter:β-glucuronidase (*GUS*) reporter gene in *N. benthamiana* leaves. Co-transformation of *FUL2_pro*:*GUS* and the empty vector pHELLSGATE8 was used as a control. **e** PT protein binding the *FUL2* promoter in an electrophoretic mobility shift assay (EMSA). A 50-bp double-stranded fragment labeled or unlabeled with 5′ 6-FAM was used as a DNA probe or DNA competitor. The FAM-labeled probe with the substitutions of four conserved bases was used as a mutant probe. The + and − symbols indicate the presence and absence, respectively, of the DNA probe or the protein. Arrows indicate the protein–DNA complex and unbound labeled DNA probe. Three independent experiments were performed. **f, g** Binding affinity of PT^H (**f**) and PT^R (**g**) for the *FUL2* promoter fragment was measured using an open surface plasmon resonance (SPR) assay. **h** Schematic illustration of the two sgRNA target sites in *FUL2*. Red lines indicate the binding sites of sgRNAs. **i** Genomic DNA sequences of *FUL2* in CR-*ful2-1*, CR-*ful2-3*, and the wild-type (TS-9). **j** Phenotype of fruit produced by the CR-*ful2* and CR-*pt^H* single mutants, the CR-*ful2* CR-*pt^H* double mutant, and the pertinent wild-type (TS-9, *PT^H* allele). All of the mutants were derived from the same wild-type background, TS-9. Source data underlying Fig. 5a, e are provided as a Source Data file.

To further characterize this interaction, we conducted transient transactivation assays with the *GUS* reporter gene in *N. benthamiana* leaves to test whether these interactions can occur in planta. Histochemical staining of the infiltrated leaves showed that the ectopic expression of *PT^H* downregulated the expression of the *GUS* reporter gene driven by the *FUL2* promoter. In contrast, high levels of GUS activity were observed in the leaves that expressed *PT^R* or that contained the empty vector (Fig. 5d). These data provided strong evidence that PT^H serves as a repressor that can downregulate the expression of *FUL2* by binding its promoter. C2H2-type zinc finger proteins specifically bind A[G/C]T core sequences[23]. Using the JASPAR database (http://jaspar.genereg.net/), we identified a DNA fragment with two adjacent A[G/C]T motifs in the *FUL2* promoter. The fragment (−1115 bp

to −1066 bp) was labeled with 5′ 6-FAM and used as a probe for EMSAs. In the EMSA, DNA-protein complexes could be detected when the 5′ 6-FAM-labeled probe was mixed with the PT^H protein. The abundance of detectable PT^H complexes was reduced when an identical DNA probe that was unlabeled or a mutant probe that contained a substitution of four conserved bases was added to the binding reaction. Notably, fewer DNA-protein complexes were detected when PT^R rather than PT^H was added to binding reactions (Fig. 5e). These results imply that PT^H directly binds *FUL2* promoter fragments with higher affinity than PT^R. To quantify the binding affinity of PT^H and PT^R for the promoter fragments, we labeled these DNA fragments with 5′ SH C6 and performed an open surface plasmon resonance (SPR) assay. By calculating the binding affinity of PT^R/H and the DNA fragment for different

concentrations of protein, we found that PT$^H$ but not PT$^R$ has a high affinity for *FUL2* promoter fragments (Fig. 5f, g). Taken together, these data indicate that PT$^H$ serves as a transcriptional repressor that binds the promoter of *FUL2* with higher affinity than PT$^R$ and negatively regulates *FUL2* expression.

To genetically characterize this interaction, we constructed a series of mutants in the same genetic background (TS-9; non-pointed tip fruit accession). We first mutated *FUL2* in TS-9 using CRISPR/Cas9 genome editing technology (Fig. 5h). Two independent *FUL2* mutants, CR-*ful2*-1 and CR-*ful2*-3, were obtained (Fig. 5i and Supplementary Data 6) that produced non-pointed tip fruit similar to the fruit produced by the wild-type (TS-9) (Fig. 5j). Then, we crossed the CR-*ful2* mutants with CR-*pt$^H$* mutants and produced CR-*ful2*/CR-*pt$^H$* double mutants. We found that the CR-*ful2*/CR-*pt$^H$* double mutants produced non-pointed tip fruit and thus, that CR-*ful2* is epistatic to CR-*pt$^H$* (Fig. 5j and Supplementary Fig. 9). Based on these data, we conclude that *PT$^H$* acts upstream of *FUL2* to control the formation of pointed tips in fruit by serving as a transcriptional repressor that downregulates the transcription of *FUL2* in tomato.

### CR-*pt$^H$* and FUL2-OE control the development of pointed tips in fruit by influencing the auxin transport

GO enrichment analysis of differentially expressed genes (DEGs) from the RNA-seq showed that a large portion of enriched terms were found to be associated with the auxin polar transport and auxin homeostasis (Supplementary Fig. 7). Previous studies revealed that auxin plays an important role in fruit development[32]. To dissect how auxin regulates the formation of pointed tips in fruit, an accession that produces fruit with pointed tips (LA4053), an accession that produces non-pointed tip fruit (TS-9), and their knockout mutants (CR-*pt$^R$* and CR-*pt$^H$*) were treated with the auxin transport inhibitor *N*-1-naphthylphthalamic acid (NPA) and plant growth regulator 4-chlorophenoxyacetic acid (PCPA), respectively. We found that the NPA treatment promoted the development of a non-pointed tip at the distal end of the fruit produced by LA4053 and their knockout mutants. In contrast, the PCPA treatment promoted the development of pointed tips in the fruit produced by TS-9. These data are consistent with auxin transport promoting the formation of pointed tips in tomato fruit (Fig. 6a–e). To investigate the potential mechanism induced by auxin to influence the development of pointed tips in fruit, we quantified the endogenous auxin content in the pointed tips of the fruit using liquid chromatography-tandem mass spectrometry (LC-MS/MS). We found that the indole-3-acetic acid (IAA) and indole-3-carboxaldehyde (ICA) content of the pointed tips from the fruits produced by the CR-*pt$^H$* and FUL2-OE plants were significantly increased relative to the wild-type (TS-9) (Fig. 6f, g). In contrast, the methyl indole-3-acetate (ME-IAA) content of the pointed tips from the fruits produced by the CR-*pt$^H$* and FUL2-OE plants was significantly decreased relative to the wild-type (TS-9) (Fig. 6h). ME-IAA is an inactive form of IAA. An esterase produces active IAA by demethylating ME-IAA[33]. Additionally, we also measured the endogenous auxin content in the pointed tip fruit from CR-*pt$^R$* lines and wild-type TS-3. We observed that the contents of IAA, ICA, and ME-IAA were similar between CR-*pt$^R$* lines and the wild-type control TS-9 (Supplementary Fig. 10). Overall, the levels of the free and active forms of auxin were significantly higher in the pointed tips of the fruit produced by the CR-*pt$^H$* and FUL2-OE plants relative to the same position in the fruit produced by the wild-type control (TS-9). To verify the RNA-seq results that suggested that auxin polar transport might be involved in the formation of pointed tips, we measured the transcript levels of the *PIN* and *AUX/LAX* genes in CR-*pt$^H$* and FUL2-OE plants. Using qRT-PCR, we found that the expression of *PIN4*, *PIN5*, *PIN9*, *PIN10*, and *LAX3* were significantly increased in the pointed tip of the fruit produced by the CR-*pt$^H$* and FUL2-OE plants relative to the same position in the fruit from wild-type (TS-9) (Fig. 6i–m). Taken

together, these data provided evidence that CR-*pt$^H$* and FUL2-OE control the formation of pointed tips in fruit by influencing the transport of auxin (see below).

### Distribution of *PT$^R$* contributed to the diversity of fruit morphology during the domestication of tomato

To determine whether *PT* could have been a target of selection in the process of tomato domestication and improvement, we evaluated the nucleotide diversity (π) between groups that we previously evaluated[30]. Comparing the ratios of nucleotide diversity between PIM and CER accessions, we found that *PT* was part of a large genomic cluster with a high ratio of diversity, indicating one or several selection events during tomato domestication (Fig. 7a, b). Additionally, as *PT* is part of a previously reported domestication sweep[34], our results suggest that the mutation/phenotype likely hitch-hiked along with other loci sweeping through. A particular candidate for selection is the *SP5G* locus[31,35]. Meanwhile, we determined the *PT$^H$* and *PT$^R$* genotypes in 311 tomato accessions (Supplementary Data 1). The frequency of the *PT$^H$* and *PT$^R$* alleles in the accessions from the wild tomato group was 98.2% and 1.8%, respectively. The frequency of *PT$^R$* gradually increased in the CER (8.6%) and BIG (24.5%) groups (Fig. 7c). We found that the *PT$^R$* (G) allele was mainly distributed in the United States, Russia, and along the Mediterranean coast (e.g., in Italy), consistent with possible hitch-hiking with other loci under selection in this large genomic region during the process of tomato domestication and later on improvement (Fig. 7d).

## Discussion

Generally, utilization of germplasms with pointed tip fruit is an effective method adopted in breeding programs to eliminate the rough blossom-end scarring in large-fruited, fresh-market tomatoes[13]. Here, we mapped and cloned a *POINTED TIP* (*PT*) gene and demonstrated that different alleles of *PT*, *PT$^H$*, and *PT$^R$*, control the development of fruit with or without a pointed tip by differentially regulating downstream genes in tomato, which provided a theoretical basis for selecting plants that produce fruit with pointed tips or smooth blossom-end scarring during breeding.

In this study, the gene controlling the fruit with the pointed tip was identified by GWAS and BSA. The lead SNP (ch05_64012700, $P = 7.57 \times 10^{-40}$) was located in an exon of *PT* and responsible for changing histidine 135 (H-135) to arginine 135 (R-135). We refer to these alleles as *PT$^H$* and *PT$^R$*, respectively (Figs. 1 and 2). Interestingly, amino acid 135 (H/R) is present in the highly conserved C2H2 domain, and the amino acid residue transition from H-135 to R-135 changes the number of C2H2 domains from three in PT$^H$ to two in PT$^R$ (Fig. 2). In general, a mutation in the coding region of a gene could result in a complete loss-of-function[11,30]. Inconsistent with previous studies, *PT$^R$* is a partial loss-of-function allele that fails to effectively suppress *FUL2* gene expression, leading to the formation of the pointed tip fruit. Thus, the formation of fruit pointed tips is due to the loss of *PT* function, which was confirmed by genetic analysis and experiments that mutated the *PT$^{H/R}$* allele in the accessions that produced fruit with non-pointed tips and pointed tips (Fig. 3). Therefore, we used *PT$^H$* mutants for our mechanical analysis of the development of fruit with pointed tips.

*PT* encodes a member of the C2H2-type zinc finger protein family that is widely distributed in plants. In *Arabidopsis*, *IDD15/SGR5*, the most homologous counterpart of tomato *PT*, together with *IDD14* and *IDD16*, regulates organ morphogenesis and gravitropism by affecting auxin biosynthesis and transport[36]. However, these INDETERMINATE DOMAIN (IDD) transcription factors contain four zinc finger domains[36,37], which is more than the 3 and 2 zinc finger domains found in PT$^H$ and PT$^R$, respectively. Phylogenetic tree analysis shows that *PT* is present in a separate group, and the most homologous gene of Arabidopsis *IDD15* is *SE3.1* (Solyc03g098070) in tomato (Supplementary Fig. 1). Previous study has shown that SE3.1 contains three zinc finger

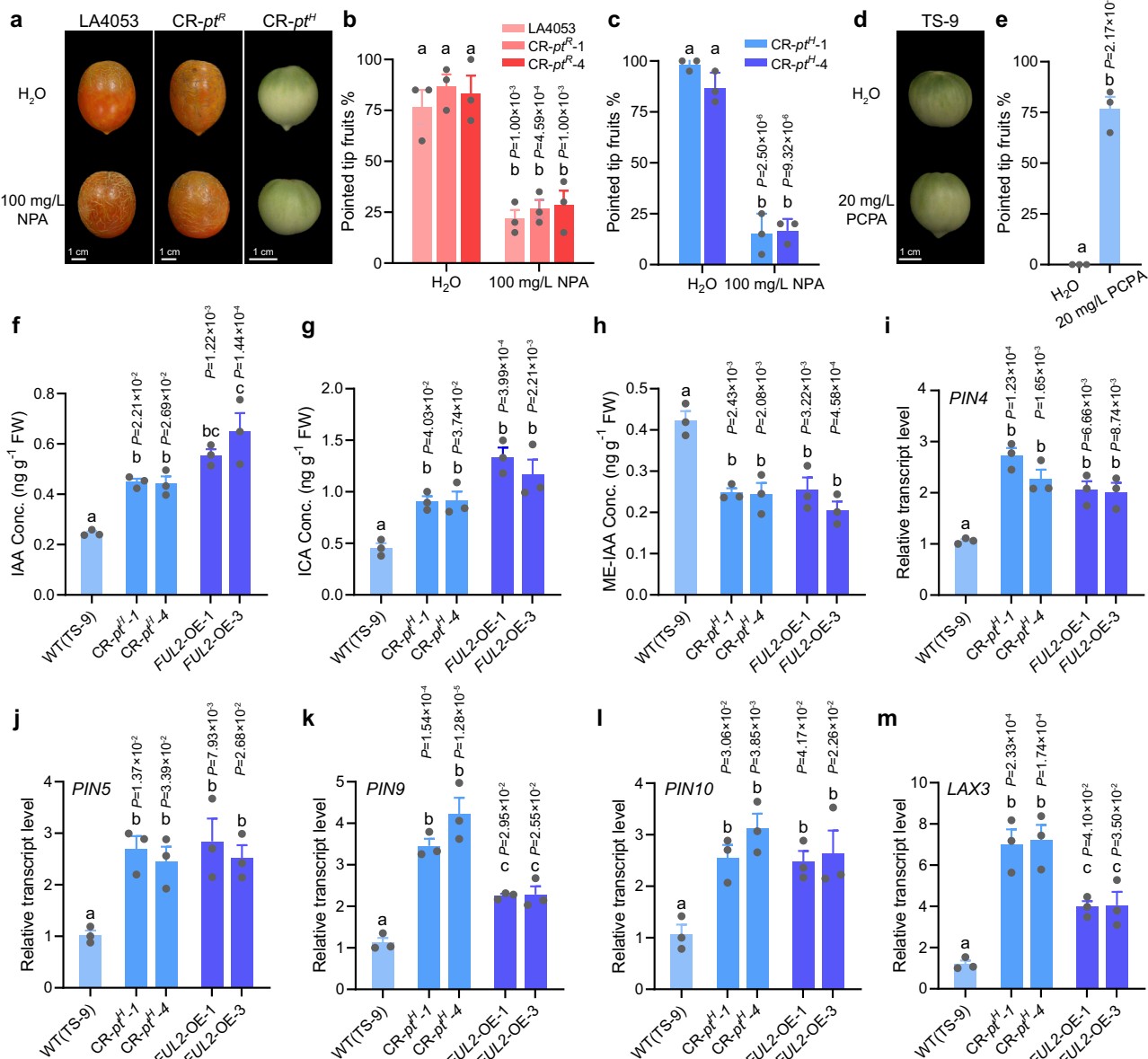

**Fig. 6 | Effects of $PT^H$ mutation and $FUL2$ overexpression on auxin distribution and the formation of fruit with pointed tips. a–c** Morphology of the distal end (**a**) and pointed tip percentages (**b, c**) of fruits from LA4053, CR-$pt^R$, and CR-$pt^H$ treated with naphthylphthalamic acid (NPA). LA4053, CR-$pt^R$, and CR-$pt^H$ produced fruit with pointed tips. **d, e** Morphology of the distal end (**d**) and pointed tip percentages (**e**) of fruits from TS-9 treated with 4-chlorophenoxyacetic acid (PCPA). TS-9 produced fruit with a non-pointed tip. The flowers during anthesis were sprayed with 100 mg/L NPA (**a**) or 20 mg/L PCPA (**d**), respectively. The treatments were alternated every day and each treatment was carried out three times. Twenty fruits from each replicate were harvested and recorded for the morphologies (pointed tip or non-pointed tip). Error bars indicate mean ± SE. $n = 3$ biological replicates.

Statistically significant differences were determined using a two-way ANOVA with Tukey's post hoc test (**b, c**) and two-tailed $t$-test (**e**). **f–h** Indole-3-acetic acid (IAA), indole-3-carboxaldehyde (ICA), and methyl indole-3-acetate (ME-IAA) content at the distal end of fruit from CR-$pt^H$, FUL2-OE, and wild-type (TS-9). The fruit was harvested at 14 DPA. **i–m** Relative expression levels of $PIN4$ (**i**), $PIN5$ (**j**), $PIN9$ (**k**), $PIN10$ (**l**), and $LAX3$ (**m**) at the distal end of fruit from CR-$pt^H$, FUL2-OE, and wild-type (TS-9). The fruit was harvested at 14 DPA. Error bars indicate mean ± SE. $n = 3$ biological replicates. Statistically significant differences were determined using one-way ANOVA with Tukey's post hoc test. Different letters indicate statistically significant differences ($P < 0.05$). Source data underlying Fig. 6b, c, e–m are provided as a Source Data file.

domains and is involved in the style extension and self-fertilization in tomato[30]. Therefore, we propose that $PT$ may have distinct functions in tomato in the regulation of the fruit pointed tip formation and some other processes, and the IDD transcription factor may present sequence and functional diversification in tomato and *Arabidopsis*. Generally, transcription factors participate in biological processes by regulating gene expression. Thus, the C2H2 zinc finger transcription factor with different numbers of conserved domains might regulate genes with different binding affinity and thus, influence the corresponding biological processes[21]. Indeed, our data indicate that the $PT^R$

and $PT^H$ alleles were involved in different biological processes. $PT^H$ is a transcriptional repressor that attenuates $FUL2$ expression by binding the $FUL2$ promoter with a higher binding affinity than $PT^R$, which inhibits the accumulation of auxin at the distal end of the fruit and the development of fruit with pointed tips (Fig. 5). To our knowledge, our report provides demonstration that different alleles of a C2H2 zinc finger transcription factor significantly contribute to fruit with pointed tips.

Intriguingly, the formation of the pointed tip is initiated after anthesis and before the style abscission (Fig. 4). The function of the

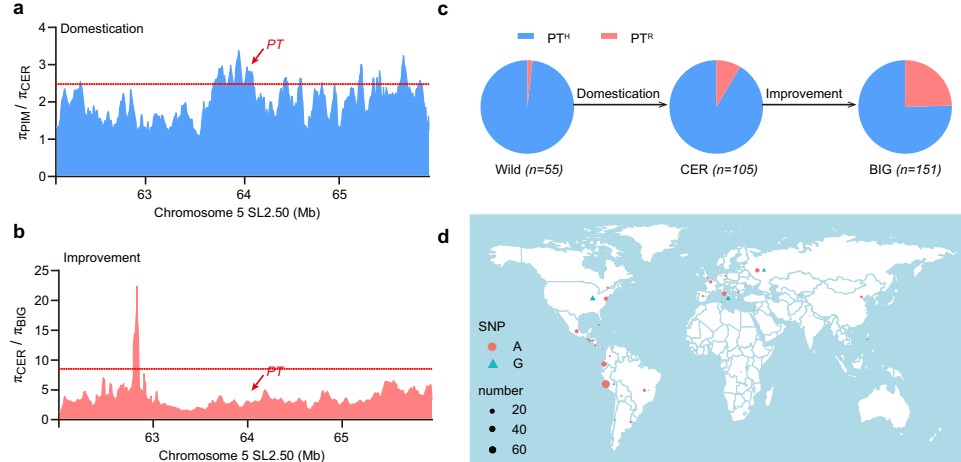

**Fig. 7 | Distribution of *PT* during domestication. a, b** Ratios of nucleotide diversity (π) surrounding the *PT* gene between PIM and CER accessions (**a**) and between CER and BIG (**b**) accessions on chromosome 5. The red horizontal dashed line indicates the top 5% threshold for the entire genome. The location of *PT* on chromosome 5 is indicated by the red arrow. **c** Frequency distribution of *PT^H* and *PT^R* alleles in wild, CER, and BIG accessions. **d** Geographical distribution of *PT^H* (A) and *PT^R* (G) alleles in the 311 tomato accessions used in this study. Made with Natural Earth. Source data underlying Fig. 7a–c are provided as a Source Data file.

style in pollination and fertilization is widely known and described in textbooks. However, when the style was detached from flowers after pollination and fertilization, the pointed tip of the fruit failed to develop in the plants of CR-*pt^H* lines (Fig. 4b). Therefore, in addition to its classic roles in pollination and fertilization, the style also plays a function to promote the initiation and growth of fruit pointed tips in plants without a functional *PT* allele (*PT^R*, CR-*pt^H*, and CR-*pt^R*). While the molecular mechanism remains unknown, it is possible that the style may transduce a signal to the ovary after pollination and fertilization in pointed tip accessions, which causes the distal cells of the ovary to divide and grow, forming the pointed tip. At the same time, the pointed tip sticks into the style like a needle, making the style difficult to fall off. This hypothesis remains to be tested further.

In addition to the *PT* locus on chromosome 5, we also identified several pointed tip loci on chromosomes 2, 3, and 7 (Supplementary Data 2). Previous genetic studies on tomato over the last several decades have discovered a group of mutants affecting the formation of the pointed blossom-end fruit, including *bk*, *n*, and *pst*, which have been mapped to chromosomes 2, 5, and 7, respectively, but have yet to be cloned[14–16]. Based on their positions on chromosomes, *PT* could be the most likely candidate for the legendary locus *nipple-tip* (*n*), whereas *bk* and *pst* may correspond to the loci mapped on chromosomes 2 and 7 in our GWAS analysis, respectively (Fig. 1 and Supplementary Data 2). Further studies are warranted to uncover the molecular nature of these loci using pertinent mutants. Interestingly, we also detected an SNP in the 5′-UTR of *FUL2* between the pointed tip and non-pointed tip fruit accessions. However, the physical position of the SNP is more than 1 Mb away from the lead SNP on chromosome 3. This may be due to the fact that the SNP variation is less frequent in these 311 tomato accessions or is not the causal one underlying the pointed tip fruit.

It is widely believed that the tomato originated in the Andes region of South America and was domesticated in Central America[4]. A recent study showed that the history of tomato domestication contains a complex and dynamic intermediate stage, and after the domestication of tomato in South America, *S. lycopersicum* var. *cerasiforme* was possibly spread northward to Mesoamerica and redomestication in Mexico[5]. The tomato was brought to Europe and distributed worldwide by the Spanish conquistadors beginning in the early 16th century. The earliest written record of tomato cultivation appeared in Italy, which described the Mandrake, a poisonous distant relative of the tomato. The fruit was recorded as flattened, segmented,

yellow, and named pomi d'oro (golden apple) in Italy[38]. Since then, different types of fruits have been documented, such as different shapes, sizes, and colors. In this study, our evolutionary analysis implied that *PT^R* could have been a target of selection during tomato domestication (Fig. 7). Coincidentally, recent research has identified the same SNP as underlying the obscure vein (obv) mutation and found that overexpression of *OBV* exhibited a change from the dark veins into a transparent phenotype[31]. Consistent with their observations, we also detected the phenotype of transparent veins in *PT^H*-overexpressing plants as compared with the wild-type TS-9. The *obv* mutant was first described as obvious variations in the vein in 1990[35]. Subsequently, several studies have suggested that the *obv* mutation may have "hitch-hiked" along with selection for mutations in the nearby *SELF PRUNING 5 G* (*SP5G*), which leads to more compact habits and shorter growing season[31,35,39,40]. Therefore, we propose that *PT/OBV*, may also have "hitch-hiked" along with the selection for mutations in the nearby *SP5G*.

Breeders usually prefer tomato fruits with non-pointed tips and smooth blossom-end scarring during the breeding process. However, the commonly used herbicides to cultivate tomatoes often contain synthetic auxins, which promote the development of fruit with pointed tips. To solve this problem, we suggest that researchers attenuate the expression of genes, such as *FUL2* and genes that contribute to auxin signaling, using CRISPR/Cas9 genome editing technology and subsequently selecting appropriate germplasm resources for breeding. On the other hand, all blossom-end morphology genes have been shown to inherit as recessives[13]. Selection of accessions with pointed tip genotypes and cross-hybridizing them with excellent inbred lines is an effective method to eliminate the rough blossom-end scarring in large-fruit, fresh-market tomatoes breeding[13]. Therefore, our study provides valuable information that can be used to select tomato germplasm that develops fruit with non-pointed tips and smooth blossom-end scarring in breeding.

In summary, we identified a tomato gene named *PT* that contributes to the development of fruit with pointed tips using GWAS and BSA. We found that an SNP located in an exon of *PT* induces an amino acid substitution that leads to changes in a C2H2 domain and generates two alleles of *PT* that we named *PT^H* and *PT^R*. Our functional complementation analysis provides compelling evidence that the loss-of-mutation in *PT^H* promotes the formation of fruit with pointed tips. Furthermore, our mechanistic analyses demonstrate that the *PT^H* allele is a transcriptional repressor of *FUL2* that binds its promoter and

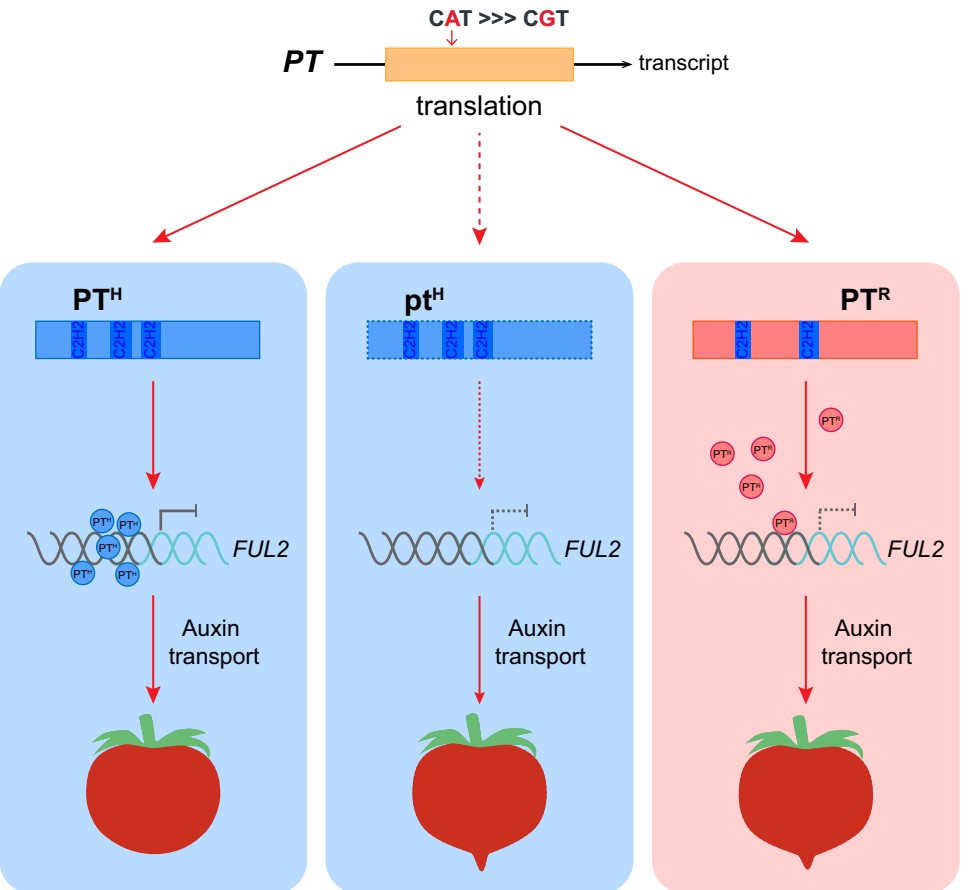

**Fig. 8 | Proposed role of *PT* in the development of fruit with pointed tips in tomato.** In this model, an SNP (A/G) in *PT* leads to the translation of either of two protein products, PT^H and PT^R. PT^H contains three C2H2 domains. PT^R has only two C2H2 domains. A knockout mutation in *PT^H* leads to the development of fruit with pointed tips. PT^H is a transcription repressor that downregulates the expression of *FUL2* by binding the *FUL2* promoter with a higher affinity than PT^R, alters the auxin transport, and inhibits the formation of fruit with pointed tips in tomato.

downregulates its expression, which affects the auxin transport and suppresses the development of pointed tips. Taken together, the variation and molecular mechanisms of different alleles that we demonstrated provide valuable information for the molecular breeding of fruit shape in tomato (Fig. 8).

## Methods

### Plant materials and growth conditions

A tomato collection consisting of 311 tomato accessions from around the world was used for the genome-wide association analysis[34]. The accessions included 1 *Solanum galapagense* (wild), 1 *Solanum neorickii* (wild), 2 *Solanum cheesmaniae* (wild), 51 *S. pimpinellifolium* (PIM, wild), 105 *S. lycopersicum* var. *cerasiforme* (CER), and 151 *S. lycopersicum* (BIG) (Fig. 1a and Supplementary Data 1). Tomato plants were grown in two independent locations, Huazhong Agriculture University and Zhongdu Seed Company[30]. For the phenotyping of these 311 accessions, we harvested the ripening fruits, observed and recorded the morphologies at the distal end of the fruit. Based on the angle of the distal end of the fruit, we classified the whole accessions into pointed tip fruits ($0° < \theta < 180°$) and non-pointed tips fruit, and we defined the protruding position of the distal end of the fruit as the pointed tip part (Fig. 1c, d). For transgenic lines and their corresponding wild-type controls, the plants were grown in a greenhouse in 16/8-h photoperiod (150 μmol m$^{-2}$ s$^{-1}$) at $25 \pm 2\,°C$ and relative humidity of 70%.

### Genome-wide association analysis

We used 4,155,820 SNPs (MAFs > 0.05, the number of varieties with the minor allele ≥ 16) to perform GWAS for the production of fruit with pointed tips with 311 accessions. The SNPs were acquired from a previous study[34]. The fruit with pointed and non-pointed tips were designated as 1 and 0, respectively, and used as phenotypic data. The association analyses were performed using the Genome-wide Efficient Mixed Model Association algorithm (GEMMA) software[41]. The first three principal components were selected as covariates. According to the number of SNPs analyzed (4,155,820), the suggestive and significance thresholds were calculated to be approximately $2.41 \times 10^{-7}$ (1/4,155,820) and $1.20 \times 10^{-8}$ (0.05/4,155,820), respectively. The partial zoom plot was constructed by LocusZoom[42]. The physical positions of the SNPs were identified using the tomato genome sequence, version SL2.50 (http://solgenomics.net).

### Linkage mapping

An F$_2$ population of 2000 individuals derived from a cross between LA4053 (a pointed tip fruit accession) and TS-9 (a non-pointed tip fruit accession) was planted in an open field at Huazhong Agricultural University. For each individual, we harvested the ripening fruits, observed and recorded the morphology (pointed tip or non-pointed tip) at the distal end of the fruit. Genomic DNA was isolated from fresh leaves using the CTAB method. For bulked segregant analysis (BSA), bulk DNA samples from the pointed tip and non-pointed tip fruit accessions were constructed by mixing equal amounts of DNA from 73 individuals that produced fruit with either pointed or non-pointed tips. Roughly 70× genome sequences for each bulk sample were generated by Novogene Bioinformatics Technology Company (Tianjin, China). The Δ(SNP-index) was obtained by subtracting the SNP-index of the pointed tip fruit bulk sample from that of the non-pointed tip fruit bulk

sample. The average Δ (SNP-index) was calculated using a 1000-kb sliding window with a step size of 200 kb. We also calculated the 95 and 99% confidence intervals of the Δ (SNP-index) using the null hypothesis of no QTLs.

## Gene expression analysis

Pointed tips from the pointed tip accessions and the corresponding pericarp from the non-pointed tip accessions were collected from the distal end of the fruit and were immediately frozen in liquid nitrogen. Total RNA was extracted using TRIzol reagent (RN0102, Aidlab, China). First-strand cDNA was synthesized using HiScript II 1st Strand cDNA Synthesis Kit (+gDNA wiper) (R212-01, Vazyme, China), as recommended by the manufacturer. Gene expression analysis was quantified using quantitative real-time PCR (qRT-PCR), which was conducted using QuantStudio™ 6 Flex System (ABI, USA). The tomato actin gene (Solyc11g008430) was used as an internal control. Primer sequences (designed using Primer Premier 5) are listed in Supplementary Data 7.

## Subcellular localization

The full-length coding sequences of $PT^R$ and $PT^H$ without their stop codons were amplified from cDNA prepared from TS-3 and TS-9, respectively, and then cloned into the expression vector pCAMBIA1302:GFP downstream of the CaMV35S promoter. Plasmids containing the $35Spro:PT^{R/H}$ and the nuclear marker $35Spro:Ghd7-CFP$ fusion genes were bombarded into protoplasts prepared from *Nicotiana benthamiana*. After incubating the protoplasts at 24 °C for 18 h, fluorescence emitted by the green fluorescent protein (GFP) and red fluorescent protein (RFP) were imaged using Leica TCS SP8 confocal laser-scanning microscope. Protoplasts harboring $35Spro:GFP$ and the fusion gene encoding the nuclear marker $35Spro:Ghd7-CFP$ were used as controls. Primer sequences are listed in Supplementary Data 7.

## Generation of transgenic tomato

The full-length coding sequence of $PT^H$ was amplified from TS-9 cDNA and then cloned into pHELLSGATE8 downstream of the CaMV35S promoter to yield the overexpression construct $35Spro:PT^H$. TS-9 and TS-3 were transformed with the $35Spro:PT^H$ transgene using *Agrobacterium* (strain C58)-mediated transformation[43]. The CRISPR-Cas9 vector pTX was used to generate $pt^{R/H}$ mutants in the TS-3 ($pt^R$), LA4053 ($pt^R$), and TS-9 ($pt^H$) backgrounds by targeting two sites in the second and third exons of $PT$ using the simple guide RNAs (sgRNA) sgRNA1 and sgRNA2 (Fig. 3e). For generating the CR-$pt^H/PT^R_{pro}:PT^R$ plants, the full-length sequence of $PT^R_{pro}:PT^R$ was amplified and cloned into pHELLSGATE8 lacking the CaMV35S promoter. The CR-$pt^H$ line was transformed with the $PT^R_{pro}:PT^R$ transgene.

To produce the $pt^H$ $ful2$ double mutant in the TS-9 background, we first used the CRISPR-Cas9 vector pTX to generate the $ful2$ mutants in the TS-9 background by targeting two sites in the fourth and seventh exons of $FUL2$ using sgRNA3 and sgRNA4 (Fig. 5h). A $ful2$ mutant was crossed to a $pt^H$ mutant. We propagated the progeny from these crosses and identified homozygous double mutants lacking the Cas9 transgene in the $T_2$ generation using PCR-based genotyping[44]. The sgRNAs were designed using CRISPR-P v2.0 (http://crispr.hzau.edu.cn/CRISPR2/). All primers and sgRNA sequences are listed in Supplementary Data 7.

## RNA sequencing

Fruits were harvested at 5 days post anthesis (DPA) from CR-$pt^H$ lines and wild-type TS-9. Pointed tips (CR-$pt^H$) and the corresponding pericarp (TS-9) were collected from the distal end of the fruit and were immediately frozen in liquid nitrogen. Samples were collected with three biological replicates and sent to Seqhealth Technology (Wuhan, China) for RNA-seq analysis. High-quality clean reads were mapped to the reference genome (version SL2.50). Differentially expressed genes (DEGs) were identified for those that had a more than twofold change in expression level with $P$ value of <0.05.

## IAA quantification assay

IAA levels were quantified in fruits that were collected at 14 days post anthesis from CR-$pt^H$-(1, 4), $FUL2$-OE-(1, 3), CR-$pt^R$-(3, 5, 6), TS-9, and TS-3. Pointed tips (pointed tip accessions) and the corresponding pericarp (non-pointed tip accessions) were collected from the distal end of the fruit and were immediately frozen in liquid nitrogen. The frozen samples were ground into a fine powder and extracted with methanol/water/formic acid (15:4:1, v/v/v). The combined extracts were evaporated until they were dry under a stream of nitrogen gas, reconstituted in 80% methanol (v/v), filtered (PTFE, 0.22 μm; Anpel), and analyzed using liquid chromatography-tandem mass spectrometry (LC-MS/MS). The auxin content of each sample (IAA: indole-3-acetic acid, ME-IAA: methyl indole-3-acetate, and ICA: indole-3-carboxaldehyde) was quantified using MetWare (http://www.metware.cn/) based on the AB Sciex QTRAP 6500 LC-MS/MS platform. Three replicates were performed for each assay.

## Yeast one-hybrid assay

The full-length coding sequences of $PT^R$ and $PT^H$ were amplified from TS-3 and TS-9, and cloned into pGADT7 (Clontech, China) using the CloneExpress II One Step Cloning Kit (C112-01, Vazyme, China). The promoter (1504 bp) of $FUL2$ was amplified from TS-9 and cloned into the pAbAi vector (Clontech, China). Following the manufacturer's protocol, *Saccharomyces cerevisiae* strain Y1HGold was transformed with pAbAi-FUL2 and cultured on an SD/-Ura medium. Subsequently, these strains were transformed with either pGADT7-PT$^R$ or pGADT7-PT$^H$ and cultured on an SD/-Ura-Leu medium. These strains were also transformed with pGADT7 to produce a strain that served as a negative control and that was cultured on an SD/-Ura-Leu medium. p53-AbAi and pGAD-Rec2-53 were used as positive control. The positive yeast strains were picked and diluted in double-distilled water. The suspension was spotted onto an SD/-Ura-Leu medium containing 50 ng mL$^{-1}$ aureobasidin (AbA) (630466, Clontech, China), or a different concentration of AbA as indicated. Primer sequences are listed in Supplementary Data 7.

## *Agrobacterium*-mediated transient expression in tobacco

The full-length coding sequences of $PT^R$ and $PT^H$ were amplified and cloned into the expression vector pHELLSGATE8. To construct a $FUL2pro:GUS$ transgene, the promoter from $FUL2$ was amplified and cloned into an expression vector (pHELLSGATE8-GUS) derived from pHELLSGATE8 that lacked the CaMV35S promoter. The primer sequences are listed in Supplementary Data 7. Using electroporation, the recombinant plasmids were introduced into *Agrobacterium tumefaciens* GV3101. We then transiently expressed the fusion proteins by infiltrating *N. benthamiana* leaves with these strains at an OD$_{600}$ value of 0.5 in *Agro*-infiltration buffer (10 mM MgCl$_2$, 10 mM MES-KOH, pH 5.7, and 150 to 200 μM acetosyringone). Two days after infiltration, the leaves were collected and incubated in a GUS staining solution.

## Electrophoretic mobility shift assay

The coding sequences of $PT^R$ and $PT^H$ that were amplified from TS-3 and TS-9 were cloned into pF3K WG (BYDV) to yield PT$^R$-pF3K and PT$^H$-pF3K, respectively. Following the manufacturer's protocol, 3 μg of PT$^R$-pF3K and PT$^H$-pF3K plasmids were added to the TnT® SP6 High-Yield Wheat Germ Protein Expression System (L3260, Promega, USA) and incubated at 25 °C for 2 h to express the proteins in vitro. EMSAs were performed using the LightShift Chemiluminescent EMSA Kit (20148, Thermo Scientific, USA). A 50-bp single-strand oligonucleotide probe from $FUL2$ was synthesized and labeled with 5' 6-FAM (Sangon, China). The same fragment without a 5' 6-FAM label was used as a competitor. The FAM-labeled probe with the substitutions of four conserved bases was used as a mutant probe. A reverse and complementary DNA probe was also synthesized and used to generate a double-stranded DNA probe. Annealing of the two DNA probes was carried out in a thermal

cycler using the following conditions: 95 °C for 2 min and then 75 °C for 30 s for the first cycle. The temperature was decreased by 1 °C for each subsequent cycle. A total of 30 cycles were performed. Then, according to the manufacturer's protocol, the protein was incubated in mixtures containing the labeled probe and different concentrations of the unlabeled competitor. Finally, the protein–DNA complexes were separated using 6% native polyacrylamide gels. The mobility of the 5′ 6-FAM-labeled probes was monitored using fluorescence (FluorChem M, ProteinSimple, USA).

## Open surface plasmon resonance analysis

PT$^R$ and PT$^H$ proteins were obtained using TnT® SP6 High-Yield Wheat Germ Protein Expression System (L3260, Promega, USA). 50-bp single-strand oligonucleotide probe of *FUL2* was synthesized and labeled with 5′ SH C6 (Sangon, China). The equilibrium-binding constant (KD) of PT protein (PT$^R$ and PT$^H$) and *FUL2* was determined by Open surface plasmon resonance (SPR)[45]. Firstly, the *FUL2-SH* probe (40 μg mL$^{-1}$) was covalently immobilized on COOH-sensor chips (Nicoya, Canada) by the 1-ethyl-3-(3-dimethylaminopropyl) carbodiimide/N-hydroxy-succi-namide (EDC/NHS) chemistry. Then, the PT$^R$ and PT$^H$ were con-tinuously diluted into several different concentrations using the running buffer and injected into the chip from low to high con-centrations. Meanwhile, bovine serum albumin (BSA) was used as a negative control. In each cycle, a 250 μL sample flowed through the chip for 5 min at a constant flow rate of 20 μL min$^{-1}$. After detection, 0.02% sodium dodecyl sulfate (SDS) was added to dissociate the peptides. Finally, the kinetic parameters of the binding reactions were calculated and analyzed using the Trace Drawer software (Ridgeview Instruments AB, Sweden).

## Statistical analysis

All statistical analyses were performed using GraphPad Prism version 8.3.0 (http://www.graphpad.com/). One-way ANOVA with Tukey's post hoc test and Dunnett's post hoc test, Two-way ANOVA with Tukey's post hoc test, and a two-tailed *t*-test were used in this study. Different letters above the bars indicate statistically significant differ-ences ($P < 0.05$).

## Reporting summary

Further information on research design is available in the Nature Research Reporting Summary linked to this article.

## Data availability

The previously reported sequence data used in this article can be found in the Sol Genomics Database [https://solgenomics.net/] with the following accession numbers: *PT* (Solyc05g054030), *FUL2* (Solyc03g114830), *PIN4* (Solyc05g008060), *PIN5* (Solyc01g068410), *PIN9* (Solyc10g078370), *PIN10* (Solyc04g056620), and *LAX3* (Solyc11g013310). The genomic sequencing datasets of BSA analysis and RNA sequencing datasets generated in this study have been deposited in Sequence Read Archive (SRA) under the accession num-ber PRJNA788715 and PRJNA861842. Other data supporting our find-ings are available within this manuscript and its Supplementary Information files. The clones and plant materials generated during this study are available from the corresponding author upon request. Source data are provided with this paper.

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

## Acknowledgements
We thank Prof. Hanhui Kuang, Prof. Qiang Xu, and Prof. Zhengming Wang for critical reading and editing of the manuscript. This work was supported by grants from the National Natural Science Foundation of China (31991182 to Z.Y., 32072595 to J.Z., and 31872118 to Z.Y.), the China Agricultural Research System (CARS-23-A13 to J.Z.), and the China Postdoctoral Science Foundation (2021M691174 to J.S.).

## Author contributions
J.S., Z.Y., and J.Z. conceived the project; J.S., C.Y., H.L., Z.Y., and J.Z. designed the experiments; J.S. performed experiments and drafted the manuscript; L.S. helped perform some of the experiments; C.L. and X.W. performed the GWAS analysis and linkage mapping; C.Z. and W.W helped plant F2 population and investigate the phenotypic data of population. G.A. carried out the geographical distribution; J.Y., C.Y., H.L., Z.H., R.M.L., Z.Y., and J.Z. supervised the project and revised the manuscript.

## Competing interests
The authors declare no competing interests.
