## [Peer Review File · Nature Communications]

Variation in the fruit development gene POINTED TIP regulates protuberance of tomato fruit tipReviewers' Comments:

Reviewer #1:

Remarks to the Author:

Review of "Two natural alleles of POINTED TIP regulate distinct processes in tomato fruit" by Song et al.

In this study the authors study the involvement of the gene Pointed Tip (PT) in the phenotype of tomato fruit with pointed tip and ripening early. They first perform a genome-wide association study (GWAS) based on 311 wild and cultivated tomatoes to pinpoint the PT gene that encodes a C2H2-type zinc finger transcription factor. Via several CRISPR modifications, they can show that a single nucleotide polymorphism changes a histidine (H) to an arginine (R) in the C2H2 domain of PT. Further molecular work by knock-out and complementation, transient co-expression in *N. benthamiana* and overexpression of each allele demonstrate that the PT_R SNP is responsible for fruits with pointed tips. It is also shown that the PT_H allele controls the development of pointed tips by downregulating the transcription of the gene FUL2, which alters the distribution of auxin. On the other hand, the allele PT_R binds to the promoter of the ACO1 gene promoting ethylene biosynthesis and fruit ripening. Overall I find the study well conducted, the statistics analyses rigorously performed and the molecular test make sense to me. However, as I am not an expert in molecular biology, I leave the other reviewers commenting on the details of the experiments. I will comment more specifically on the GWAs and evolutionary analysis on the gene.

I do not have major reservations on the work and suggest only two main changes.

1) While I do not doubt the GWAs results, it is unclear whether the authors have controlled for genetic relatedness (population sub-structure) in their GWAs analysis. It is usual to, for example, take into account some PCA components as covariates into the GWAs. As the authors include wild and cultivated species, the GWAs results could be biased by phylogenetic relationships and/or relatedness and substructure in the cultivated sample. Could the author explicit this or justify the assumptions underlying the GWAs analysis?

2) Regarding the evolutionary analysis to look for selection at the PT gene, I am left unconvinced by the evidence. Basically, while it is possible that selection acted on the PT gene, the results shown in Figure 7 can also be explained by hitch-hiking of this gene with another one in proximity which is the real target of selection. As mentioned by the authors the PT gene is part of a sweep region identified in the Lin et al. study, however this region may be large and contains many genes (with high linkage disequilibrium). At that point the authors do not need to demonstrate selection for this current study to hold up so I would suggest the following options:

- tone down the claim on selection acting at the PT gene and say that it "could potentially" be under selection but they cannot fully demonstrate it.
- do some additionally work to present a better proof of selection by investigating the patterns of selection at neighbouring genes to account for the effect of linkage disequilibrium. Also it would be then a good idea to use new data from the Razifar et al. 2020 paper (see below) who provide more sequences of *S. lycopersiforme* and *S. lycopersicum* from meso-america. This requires some additional work on performing sliding windows tests of selection across genes around the PT gene and test for LD correlations.

Minor comments:

- In the introduction and discussion, the Razifar et al. paper (Mol Biol Evol 2020) should be cited and used as it provides more up to date scenario for tomato domestication.
- *Solanum pimpinellifolium* is a wild species and cannot be used as cultivated in the analyses (the authors should clarify that this was not the case).
- line 121: "are" instead of "is".
- in the result section it would be nice to define again PT_R and PT_H as a reminder for the reader.
- lines 466-490: I found this part of the discussion to be very speculative and fuzzy. The authors

should back up their assumptions with references or tone down (or remove some claims). For example the part from line 476-484 seems highly speculative and I am not sure that I agree with these arguments. Is it useful to speculate on the future use of the PT alleles in breeding without providing more solid arguments?

Reviewer #2:

Remarks to the Author:

The manuscript submitted by Song et al. describes the identification of a novel locus (PT) controlling the formation of pointed-tip at the distal part of tomato fruit via the implementation of a GWAS approach and then through reverse genetics strategies (CRISPR knock-out and overexpression). The authors present convincing data showing that a C2H2 transcription factor (named here PT) is involved in determining round or pointed-tip formation. In particular, the data revealed that a single nucleotide change (G->A) in the second C2H2 domain of the PT transcription factor gene generates two alleles of PT (PTH and PTR) that promote the transition from a round shape to a pointed tip via altering of the affinity of this TF to downstream promoter genes. The authors also show that the PTR locus was selected during the tomato domestication process, since wild tomato accessions do not produce fruit with pointed tips, while more than 21 % of the *S. lycopersicum* BIG accessions exhibit fruit with pointed tips. Using yeast one-hybrid, transient transactivation assay, EMSA and DNA-protein binding affinity assay, the authors reveal that PT can repress transcription of the FUL2 gene by direct binding to its promoter. Moreover, they show that the PT/FUL2 regulatory module controls auxin transport and distribution, therefore determining the development of round tomato or pointed tips.

Although the manuscript does provide real novelty in identifying a fruit tip locus that has not been previously described, the study fails, however, to provide any histological information on how and when the fruit tip is initiated. More importantly, the manuscript contains some inconsistent data and a number of misinterpretations of the results. Also, the experimental design is not always optimal. Surprisingly, comparative global transcriptomic profiling is missing in this study whereas RNA-seq analysis could have provided insight into the gene networks underlying tip formation. The data related to the regulation of fruit ripening are confusing and weak. Actually, they are irrelevant to the scope of the paper and it would be rather beneficial to discard the part dealing with fruit ripening from the manuscript.

The pointed tip formation is an interesting topic when aiming to further understand the factors underlying the wide diversity of fruit morphology, but the agricultural importance of this trait has not been addressed in the manuscript which reduces the overall impact of the paper.

The manuscript suffers from several serious shortcomings that need to be thoroughly addressed.

Major comments:

1. A broad introduction of the formation fruit tip in tomato and in other fruit species is missing. What is known about the organ/tissue from which the fruit tips are formed and when this process starts to take place?
2. It is rather surprising that the GWAS didn't detect SNPs associated with other genes (those already known) involved in the pointed tip trait. What about also the detection of SNPs in the FUL2 gene?
3. Is there a particular reason why accession LA4053 was chosen for the cross between round and pointed tip, as this netted skin line seems to have very strong and peculiar phenotypes?
4. It is not clear how the pointed tip trait is defined. This definition is lacking in both methods and legend to figures. When conducting the RT-PCR experiments and IAA quantification assays, there is no description on how the samples were collected from round and pointed-tip fruit?
5. The expression at the transcript level is only provided in one accession, what about other genetic backgrounds? Also, at which stage of tip development PT is expressed? Expression analysis of TP gene at different stages spanning the tip formation period is missing.
6. The expression patterns assessed by qRT-PCR and by GUS reporter fused to PT promoter do not seem to match. What is the length of the promoter used to drive the GUS reporter gene? Does it

include the 5'UTR?

7. Since PT is expressed in several tissues/organs, it is surprising that the KO mutation doesn't result in more phenotypes in addition to pointed tip. How many PT orthologs are in the tomato genome? Could it be that the absence of additional phenotypes is due to functional redundancy among homologs. A phylogenetic analysis of PT genes is missing. Also, a presentation of the C2H2 family of TFs and their function in the tomato and other plant species is missing in the introduction section.

8. It is surprising that PTR-OE lines exhibit pointed tips since it is shown that the PTR allele is recessive to PTH.

9. The claim that PTR promotes the formation of fruit with pointed tips and that PTH allows development of round fruit is an overstatement. It seems that the pointed tip trait is dependent on the absence of the PTH allele, regardless of whether or not the PTR allele is present. This is supported by the phenotypes CR-PTH-KO lines. Again, this makes highly questionable the presence of the pointed tip phenotype in the TS-9 lines expressing the PTR transgene.

10. It is rather confusing that the authors did not use the same genetic background of round and pointed tip accessions for the generation of the different mutants. Using the same genetic background all over the study would have facilitated the comparison of the impact of all mutations (overexpression, knockout). When trying to decipher the functional significance of PT, going from the LA4053 accession, to TS-3 and then to Micro-Tom for the pointed-tip lines makes the interpretation of the data more difficult as they all display variable traits in addition to the pointed tip.

11. Figure 3B shows clear pointed tips in PTR-OE lines, but the transcript level of FUL2 (Fig. 4A) was not altered. This is inconsistent with the pointed-tip being dependent on high expression of FUL2. In my opinion, OE-PTR lines are not the good control to use for evaluating the function of PTR and for addressing its physiological significance in comparison to that of its PTH allelic variant. Instead of overexpressing 35S-PTH/R in PTH genetic background, it would be more informative to express proPT:PTH/R in the PTH-KO or PTR-KO lines.

12. Figure 4E, 4F and 4G are not convincing in showing the binding specificity of PTH compared to PTR.

13. In order to demonstrate the epistatic relationships between PT and FUL2 genes, I am not sure that the most appropriate genetic experiments are those reported (Figure 4J). The CR-ful2/CR-ptH double mutant is not meaningful in this regard. More convincing would be to express the 35S::FUL2 gene in the 35S::PTH genetic background as in this way FUL2 should escape the PTH-dependent down-regulation. Therefore, if PTH is really acting via down-regulation of FUL2, then these lines should display the pointed tip phenotype despite the presence of a functionally active PTH protein.

14. In Figure 5A and 5B, there is no description on how the NPA and PCPA treatments have been performed. Did the authors perform injection of the flower or the fruit? At which stage? How many times?

15. The DR5:GUS expression data presented in Figure 5C are not convincing. It is not obvious that the blue staining is more intense in CR-ptH than in TS-9. This is not surprising since, to my understanding, the crossing gives heterozygous lines in which the PTH gene is still able to produce enough amount of the corresponding protein to repress the expression of FUL2 gene.

16. Figure 5A and 5B show that tip formation might be associated with auxin transport, but they are using different genetic background. A more solid way would be to perform NPA and PCPA treatment directly in PTH/R OE and KO lines.

17. In all figures showing fruit pictures, there is no indication on how many fruits are being tested and what is the percentage of fruits showing round or pear shape with/without pointed-tips. Some figures lack statistical analysis. For example, Fig.4E and Fig. 5A-C. In Fig 5C, it is difficult to conclude on whether the spatial expression patterns of DR5-GUS are totally different among the three lines or whether they only differ with regard to the expression in the distal part of the fruit. In most figures, it is not stated how many fruits from each line were used to conduct the experiments?

18. The early ripening story is difficult to understand because it defies the principles of what is widely accepted regarding the mechanisms of climacteric fruit ripening. The high expression of ACO1 at 24 dpa (Figure 6A) can hardly be correlated with the early ripening. At this early immature green stage, the fruit didn't gain their competence to ripen. Moreover, the data in Figure 6A and Figure 6B are not consistent and ACO1 expression unlikely reflects the early ripening. Indeed, it is shown in the Figure

that at 34 dpa ACO1 expression is similar in WT and PTR-OE fruit while at the same stage (34 dpa) ethylene production is 5 times higher in PTR-OE. Does similar enhanced ethylene production occur also in CR-ptH fruit? Surprisingly, Figure 3L seems to indicate that ptH-KO lines don't display early ripening while they are anticipated to produce more FUL2 proteins like PTR-OE lines. Do the tomato accessions bearing the PTR allele show faster fruit ripening than those with PTH allele?

19. Figure 6F shows that PTR but not PTH is able to bind the DNA fragment containing the A(G/C)T motif, while Figure 4E shows that PTH binds the A(G/C)T containing fragment with higher affinity than PTR. This is fully inconsistent.

Keeping with the same idea, Y1H shows that PTR doesn't bind to FUL2 promoter (Figure 4C), but this is inconsistent with the outcome of the EMSA experiments showing that PTR binds to FUL2 promoter fragment (Figure 4E).

Additional comments:

1. It is not true that all fruit develop from the ovary (Line 62). Strawberries for example develop from the receptacle portion of the stem of the flower.
2. How to define the pointed tip trait? To which degree of the angle formed at the distal end of tomato fruit it is called pointed-tip? This definition is lacking in both methods and figures. Round fruit and pointed-tip are two different traits. Although most round fruits are shown without tips, I think the author could change "round fruit" to "no tips" as this is how they conducted the GWAS study.
3. In Methods section, when conducting the RT-PCR experiments and IAA quantification assay, how were the pointed-tips collected from fruits, especially those materials giving round fruit. It is not clearly explained in the manuscript.
4. The title of Figure 2 is not correct. "Loss of one C2H2 domain in PTR from an adenine (A) to guanine (G) mutation in PT". Similarly, the title of Figure 3 is confusing "Loss-of-mutations in both PTR and PTH alleles that led to the formation of fruit with pointed tips". It is not clear what loss-of-mutation means? The titles of most figures are not well written and need editing.
5. The CR lines generated for PT, FUL2 and ACO1 genes in the three tomato accessions are not well represented in Fig. 3I-K, Fig. 4I and Fig. 6I. Are they all KO lines? Which regions are represented for the C2H2 or other conserved domains? The predicted protein from CDS is not indicated in the figures.
6. Does PT expression at the protein level is similar in round and pointed tip mutant? It would be more correct to say that the expression is assessed at the transcript level only.

Reviewer #3:

Remarks to the Author:

Fruit shape defined by several features of the fruits is an important aspect of fruit quality and consumer acceptance by and large. Some tomato varieties, especially cherry tomato, produce pointed fruits. Though auxin over-production has been linked to outgrowth of the distal end of the fruit, causing the formation of pointed fruits, the genetic basis of this trait is not known. Song et al in this paper report the identification of natural variants associated with the formation of pointed fruits in tomato. Using 311 tomato accessions previously re-sequenced, they by GWAS analysis identified a SNP in the coding region of Solyc05g054030 highly associated with the pointedness of the distal fruits. An elegant phenotypic analysis of loss-of-function mutants and overexpression lines of Solyc05g054030 in different genetic backgrounds confirmed the H135R mutation is response for the formation of the pointed tips. The authors hypothesized that the MADS gene FUL2 is a potential PT target based on a similar phenotype they observed on FUL overexpression lines instead of performing a genome-wide identification of PT targets. Fortunately, the authors have been able to show that the zinc finger protein PT binds to FUL promoter by several in vitro assays and in *N. benthamiana* leaves, further supported by a genetic analysis using pt ful double mutant. As previous work has shown that pointed tips often happen in the fruits over-producing auxin, the authors confirm the formation of the pointed tips controlled by PT is also attributed to high auxin signals in the distal ends of the fruits.

Overall, this is an interesting story, though the significance of these findings is arguable.

Despite PT is within a domestication sweep and the PT (R) allele is prevailing in large fruit accessions, it is debatable that the trait was targeted during domestication. Pointed fruits apparently have no contribution to the adaptation to growth conditions or nutrient improvement in tomato. More likely the H135R mutation was rose and somehow retained in cultivated tomato, unless there are unknown traits regulated by PT that was targeted by domestication.

Solyc05g054030/PT from the sequenced reference genome of Heinz1706 also contains the R135 variant like those from TS-3 and LA4053, but Heinz1706 set fruits without pointed tips. This suggests that the R135 variant in this gene is not solely response for the phenotype.

Solyc05g054030/PT shares high sequence similarity with Arabidopsis IDD15/SGR5 that together with IDD14 and IDD16 regulates auxin biosynthesis and transport and is involved in organ morphogenesis and gravitropic responses in Arabidopsis. In tomato, it may have different functions, but these INDETERMINATE(ID)-DOMAIN proteins (IDDs) have four zinc finger domains (ZF1-ZF4) (Colasanti J, Tremblay R, Wong AY, Coneva V, Kozaki A, Mable BK. The maize INDETERMINATE1 flowering time regulator defines a highly conserved zinc finger protein family in higher plants. BMC Genomics. 2006 7:158). Does PT truly contain two ZF domains? Discussion on the functional diversification/divergence between PT and IDDs from other species is warranted.

FUL2 (and FUL1) has been shown to regulate fruit ripening in tomato (Bemer M et al., 2012 Plant Cell 24(11):4437-51; Fujisawa M et al., 2014 Plant Cell 26(1):89-101). If PT targets FUL2 as shown in Fig 4, early ripening observed in fruits overexpressing the PT (R) allele may be caused by disrupted FUL2 activity. The possibility may need further investigation.

Reviewer #4:

Remarks to the Author:

The authors describe how they by a combination of GWAS and map-based cloning, identified a single SNP in the coding region of a C2H2 Zinc finger transcription factor that is responsible for the presence/absence of a pointed tip on tomato fruit. The "wild" allele appears to code for three Zn fingers, while the derived allele contains a disruption in one of the three, apparently causing the pointed tip. They then show that the full 3 Zinc finger version likely suppresses FUL2 expression by direct interaction with the promoter and that interference with that process causes a local increase in auxin, in turn causing the tip phenotype. Conversely, the derived allele encodes a protein not able to suppress FUL2 but in turn activating ACO1, ethylene production, and early ripening (Fig. 8). The molecular characterization looks good but contains a few gaps that I would like to see resolved (see below).

Furthermore, as good as the molecular characterization may be, a good developmental basis for the phenotype is lacking, which is a pity. What particular developmental process causes the tip phenotype? Is it determined already in the ovary? Or during what stage? What does the tip consist of? Local increase in cell division or expansion? Only a single figure (1J) is shown without further elaboration. Furthermore the phenotyping for "pointed tip" lacks a quantitative description. When is a tip considered pointed? In Fig. 1C, the difference between Ts-9 and most of the others seems obvious, but TS-246? Most pointed fruits are also elongated. Is there a link? Fig. 1K seems to indicate that there at least two more loci involved.

In their discussion, the authors suggest that pointed tips may have been selected for during domestication due to their consumer appeal? I find that hard to believe & did the authors check whether the pointed tip accessions are really consumer tomatoes rather than processing ones? The authors are forgiven for not noticing that last year, probably during the preparation of their

manuscript, another group identified exactly the same SNP as underlying the "obscure vein" (obv) mutation (Lu, J. et al. (2021). OBV (obscure vein), a C2H2 zinc finger transcription factor, positively regulates chloroplast development and bundle sheath extension formation in tomato (*Solanum lycopersicum*) leaf veins. *Hortic. Res.* 8.). Just as those authors do not mention the pointed tip phenotype, in this study surely the obv phenotype must have been apparent? It would be good to discuss this. It was suggested that the obv mutation has "hitchhiked" along with selection for mutations in the nearby SP5G which leads to more compact plant habit and shorter growing season. For PT, the same may very well apply, more likely than consumer preference.

Specific comments:

Title: is not cover the contents very well

line 78: what is "fruit flatness" Did you mean surface smoothness (as opposed to ribbed or blocky)?

line 143: "raised" requires a quantitative measure

line 203-204: PT is highly expressed in leaves, flowers, and styles, but much lower in ovary and immature fruit; this is also relevant for the question where and when PT action takes place. The GUS activity of the promoter fusion (fig. S5B is not very clear, but seems to show that expression is particularly high in the distal end of the ovary. Expression of PT could have been determined for the distal part of the immature fruit to see whether it is locally high there vs. other parts of the fruit.

Overexpression of FUL2 causes delayed style abscission (Wang et al). Is that the case in PT(R) fruits as well. What is the relation with the pointed tip? See also line 257

Line 312: Why Micro-Tom? in my experience MT has only pointed tips some of the time (mostly correlated with parthenocarpic development), not always. Does it contain the pt SNP?

Line 348 and onwards: PT(R) overexpression is not a good proxy for determining its function. This should be observed in isogenic lines with either of the two alleles and confirmed by knockout in the PT(R) lines.

Lin 458-460 seems to be wrong

Reviewer #5:

Remarks to the Author:

The manuscript entitled "Two natural alleles of POINTED TIP regulate distinct processes in tomato fruit" describes the mapping and cloning of PT by GWAS, biparental mapping and plant transformations. The two alleles are reported to differ in protein sequence resulting in a C2H2 protein with the predicted 2 or 3 zinc finger domains. The difference in two versus three domains leads to a dramatic difference in efficiency of promoter binding to either FUL2 or ACO1, respectively, in a Y1H and EMSA. One allele PTR, the zinc finger appears to function as a transcriptional activator whereas for the other allele, PTH, the protein appears to function as a transcriptional repressor as measured in tobacco cells or leaves. The authors show a link to auxin transport for PTH, leading to round fruits. The allele producing pointed fruits, PTR, is effectively a loss-of-function allele of PTH with respect to bottom fruit shape. Loss of function of PTH also leads to earlier ripening (see figure 3L), suggesting that this locus controls ripening as well. The role of PTR if any, is less clear in the plant. Whereas overexpression of PTR leads to earlier ripening, the converse, a loss of function of PTR does not lead to delayed ripening (fig 3M). This leads this reviewer to assume that PTR in the plant is likely a loss of function for both bottom fruit shape as well as ripening, even though molecular studies such as promoter binding in gel shifts and Y1H support a role in transcriptional activation.

Regardless, the study is quite interesting, and many parts of the research are solid with additional evidence to support the interpretation of the findings (figs 1-2). Other parts could be improved by additional experimentation to derive at the most plausible explanation. Fig 3B shows that overexpression of PTR overrides wild type function of PTR presumably by creating a dominant negative complex or a poisoned interaction of the two alleles. This could explain the observed pointed shape and early ripening in 3B which is the most likely interpretation when taking the whole of figure 3 into account. Yet, the authors do not consider this as a reasonable explanation of the phenotype observed

in Fig 3 and instead assume that PTR plays an active role in regulating ripening. As mentioned above, if PTR was actively regulating ripening, the KO of this allele (Fig. 3M) would have resulted in delayed ripening. The interpretation of phenotypes observed by overexpression should always be taken with caution. The creation of a pear or oval shaped fruit was only found in lines overexpressing the PTH allele, further supporting that this allele and not PTR is the only functional allele. Therefore, the downstream molecular experiments, specifically the ACO1 promoter analyses, are meaningless if this gene is not affected by expression of PTR in vivo. This situation can easily be clarified if the authors were to conduct RNA seq with the near isogenic lines (from Fig 3A but ideally more inbred than an F2:F3), as well as with the CR and OX of PTH and PTR lines. Moreover, RNA seq would also need to be conducted in order to validate the notion that "only" auxin and "only" ethylene regulate pointed tip and ripening respectively. The last part of the study regarding whether PTR was selected during domestication is weak. If the PTR was selected during domestication, the allele should be fixed in modern (aka BIG) tomato and it is not. I would remove the last figure and instead only focus on the most interesting evolutionary aspect of the phenotype is that many pointed shaped tomato varieties carry this PTR allele. Lastly, the introduction nicely introduces the known loci controlling pointed shape in tomato, such as bk, bk2, n and pst, the discussion never relates these known mutations to the PT locus. While these old mutants have been around forever, the underlying genes have not been identified. Therefore, this study is potentially clarifying the molecular underpinnings of mutations that have been described 100 yrs ago. Thus, is PT corresponding to n? nipple tip? It looks like that bk and pst correspond to the other loci mapped in figure 2. The authors should clarify which of the known loci PT corresponds to.

Specific comments are below:

- In line 128, the authors refer to the "nucleotide substitution from G to A" is incorrect. A is wildtype and therefore the substitution is A to G.
- Line 146, replace high for many.
- Line 148-149. The minor allele frequency (MAF) is a frequency. If 5 out of 311 is the cut off, that is a MAF of $5/311 = 0.016$. Please reanalyze the data with the correct MAF of 0.05 (which is the best MAF) which is 16 or more.
- Make sure that if referring to recessive nature of the trait (pointed tip) the authors back that up with showing the number of plants with tip and number of plants with round fruit (and intermediate if they were observed as well). To confirm the recessive nature of PTR, please present the number of plants that are homozygous PTH, homozygous PTR and heterozygous with the phenotype.
- Line 159, replace speculated with hypothesized.
- Line 180, the gene mutation is predicted to lead to a protein that has fewer C2H2 domains. Replace the word "alter" to "is predicted to alter"
- Line 185 and 187, reverse the statement changes an arginine to a histidine residue. Histidine is wild type and thus the mutation changes the histidine to an arginine.
- Line 198, this paragraph is not starting appropriately. The first sentence should be part of the last paragraph ending on line 197. In addition, add that the promoters of the alleles of the parents in the biparental cross were sequenced and not the whole GWAS population.
- Line 204, remove "In fact, PT suppl figure S5A) because it is not necessary.
- Remove the GUS data is figure S5B because the data are not convincing. Same for Fig 5C. The GUS expression pattern is weak and diffuse and not specific to the tip.
- Line 208, "taken together..." is not correct. The expression of GUS or PT does not show a change in protein sequence. Please modify the sentence accordingly.
- Line 227, the shape of the fruits is more oval than pear. I would consider that overexpression of PTH leads to pear or oval fruit depending on the genetic background. Or oval in both cases.
- Line 237, Specifically, PTR promotes the formation of fruits with pointed tips is likely incorrect (see comments above). It is not likely based on the data shown that PTR has a function in the plant.
- Line 242, add the word endogenous before PT.
- Line 247, "... that changed the reading frame.." is not correct. One mutant is a 9 bp deletion and that is an inframe mutation. The authors need to explain how an in frame mutation has the same

affect as an out of frame mutation. Moreover, the authors should present in the supplement the resulting amino acid sequence in the CR lines for each allele. In one case, the gRNA seems almost in the intron, does it affect amino acid sequences or not?

- Line 254, the overexpression of PTH leads to an oval shaped fruit. The term round edge at the distal end is ambiguous.
- Fig 4E would benefit from using mutant probe sequence as well to determine if that can not compete with the wild type probe.
- Line 287, replace indicate with imply because the statement is too strong.
- Line 315, rephrase the rounded edge terms as they are ambiguous.
- Line 318, add transport after auxin.
- The manuscript has several transitions that are odd. The transition from phenotype to FUL2 and auxin is unclear. Yes, overexpression of FUL2 leads to pointed fruit shape and yes auxin controls tons of processes in plants including pointed fruit tips. Yet, focusing only on auxin transport (PINs and LAX) does not mean that auxin is controlling these processes exclusively because other processes were not ruled out. The unique role of auxin and FUL2 would be more convincing if they were identified through an RNA seq experiments. The authors should show that GA or ethylene (for example) are not involved by qRT PCR or by RNA seq.
- The role of auxin in regulating fruit shape could be strengthened by demonstrating that TS3 pTR does not affect IAA levels.
- Lines 338-347, since the expression of only some genes is evaluated, it is not possible to conclude that auxin transport is critical because the authors have not demonstrated that other processes and hormone genes are not altered.
- Line 348, Similarly the transition to ethylene is not strong. For example, why focus on ACO1 and not other genes?
- Fig 6B should show ethylene accumulation at an earlier and same time point as in A, 24 dpa. The ethylene accumulation experiment should also be conducted in the NILs.
- Explain in fig 3H the meaning of the black dotted line.
- Figure 3, For PT's role in ripening, the authors should present a controlled fruit ripening experiment. Namely, fruits need to be tagged and ripening in dpa should be recorded in all transgenic lines as well as the NILs.
- The dpa should be shown in all figure panels that show images of fruit at different maturation stages.
- The legends for most figures could be improved (Representative pieces of fruit, for example, is incorrect phrasing).
- Figure 5, the role of auxin inhibitors would be much better studied in a normal tomato such as the parents of the mapping study. Microtom has hormone mutations to accommodate its compact growth and these could interact with auxin in regulating the pointed phenotype. Does Microtom carry the PTR allele? In any case, the authors should repeat the experiment with the lines used in the mapping population.

Point-by-point responses to Reviewers' comments

Reviewer #1 (Remarks to the Author):

Review of "Two natural alleles of *POINTED TIP* regulate distinct processes in tomato fruit" by Song et al. In this study the authors study the involvement of the gene *Pointed Tip (PT)* in the phenotype of tomato fruit with pointed tip and ripening early. They first perform a genome-wide association study (GWAS) based on 311 wild and cultivated tomatoes to pinpoint the *PT* gene that encodes a C2H2-type zinc finger transcription factor. Via several CRISPR modifications, they can show that a single nucleotide polymorphism changes a histidine (H) to an arginine (R) in the C2H2 domain of PT. Further molecular work by knock-out and complementation, transient co-expression in *N. benthamiana* and overexpression of each allele demonstrate that the *PT_R* SNP is responsible for fruits with pointed tips. It is also shown that the *PT_H* allele controls the development of pointed tips by downregulating the transcription of the gene *FUL2*, which alters the distribution of auxin. On the other hand, the allele *PT_R* binds to the promoter of the *ACO1* gene promoting ethylene biosynthesis and fruit ripening. Overall I find the study well conducted, the statistics analyses rigorously performed and the molecular test make sense to me. However, as I am not an expert in molecular biology, I leave the other reviewers commenting on the details of the experiments. I will comment more specifically on the GWAs and evolutionary analysis on the gene.

Reply: We thank the reviewer for the concise summary and support for this research.

I do not have major reservations on the work and suggest only two main changes.

1) While I do not doubt the GWAs results, it is unclear whether the authors have controlled for genetic relatedness (population sub-structure) in their GWAs analysis. It is usual to, for example, take into account some PCA components as covariates into the GWAs. As the authors include wild and cultivated species, the GWAs results could be biased by phylogenetic relationships and/or relatedness and substructure in the cultivated sample. Could the author explicit this or justify the assumptions underlying the GWAs analysis?

Re: We are sorry for our carelessness that we missed the covariates information. We had taken into account three PCA components as covariates in the GWAS analysis. Additionally, the BSA analysis from the F2 population also identified the same fruit tip locus. Thus, we believe that the conclusion on the mapping region of the *PT* gene is reliable. We provided this information in the Materials and Methods section of the revised manuscript on lines 543 to 544.

2) Regarding the evolutionary analysis to look for selection at the *PT* gene, I am left unconvinced by the evidence. Basically, while it is possible that selection acted on the *PT* gene, the results shown in Figure 7 can also be explained by hitch-hiking of this gene with another one in proximity which is the real target of selection. As mentioned by the authors the *PT* gene is part of a sweep region identified in the Lin et al. study, however this region may be large and contains many genes (with high linkage disequilibrium). At that point the authors do not need to demonstrate selection for this current study to hold up so I would suggest the following options:

- tone down the claim on selection acting at the *PT* gene and say that it "could potentially" be under selection but they cannot fully demonstrate it.

- do some additionally work to present a better proof of selection by investigating the patterns of selection at neighbouring genes to account for the effect of linkage disequilibrium. Also it would be then a good idea to use new data from the Razifar et al. 2020 paper (see below) who provide more sequences of *S. lycopersiforme* and *S. lycopersicum* from meso-america. This requires some additional work on performing sliding windows tests of selection across genes around the *PT* gene and test for LD correlations.

Re: We appreciate for the suggestions. Indeed, there is room for improvement on the evolutionary analysis of the *PT* gene reported in this work. We also agree that *PT* could have hitchhiked an unidentified, cryptic locus in the close proximity, which is the real target of selection. Based on the comments of Reviewer #4, we found that we ignored a newly-reported study published while we were preparing our manuscript. They identified the same SNP underlying the "*obscure vein*" (*obv*) mutation¹. Previous studies suggested that *obv* mutation has "hitchhiked" along with selection for mutations in the nearby *SP5G* which leads to more compact plant habits and shorter growing season^{1,2,3}. Therefore, *PT*, namely *OBV*, also has "hitchhiked" along with selection for mutations in the nearby *SP5G*. Thus, we tone down the claim with "could potentially" under selection in the revised Result section (lines 378 and 389) and provided this "hitchhiked" information in the Discussion section of the revised manuscript on lines 480 to 492.

Minor comments:

- In the introduction and discussion, the Razifar et al. paper (Mol Biol Evol 2020) should be cited and used as it provides more up to date scenario for tomato domestication.

Re: Thank you for the suggestion. The Razifard et al. paper (Mol Biol Evol 2020) reveals that the history of tomato domestication contains a complex and dynamic intermediate stage, and after the domestication of tomato in South America, *S. lycopersicum* var. *cerasiforme* was possibly spread northward to Mesoamerica and redomestication in Mexico⁴. We added this information in the revised manuscript on lines 41 to 44 and 471 to 474.

- *Solanum pimpinellifolium* is a wild species and cannot be used as cultivated in the analyses (the authors should clarify that this was not the case).

Re: We apologize for the ambiguous description. In our study, *Solanum pimpinellifolium* was used as a wild species. We annotated it after PIM in the revised manuscript on line 131.

- line 121: "are" instead of "is".

Re: It is corrected as suggested.

- in the result section it would be nice to define again *PT_R* and *PT_H* as a reminder for the reader.

Re: As suggested, the "*PT_R* and *PT_H*" are defined again in the Results section of the revised manuscript on line 175.

- lines 466-490: I found this part of the discussion to be very speculative and fuzzy. The authors should back up their assumptions with references or tone down (or remove some claims). For example the part

from line 476-484 seems highly speculative and I am not sure that I agree with these arguments. Is it useful to speculate on the future use of the PT alleles in breeding without providing more solid arguments?

Re: The speculative statements are deleted in the revised manuscript.

Reviewer #2 (Remarks to the Author):

The manuscript submitted by Song et al. describes the identification of a novel locus (*PT*) controlling the formation of pointed-tip at the distal part of tomato fruit via the implementation of a GWAS approach and then through reverse genetics strategies (CRISPR knock-out and overexpression). The authors present convincing data showing that a C2H2 transcription factor (named here PT) is involved in determining round or pointed-tip formation. In particular, the data revealed that a single nucleotide change (G->A) in the second C2H2 domain of the PT transcription factor gene generates two alleles of *PT* (*PT^H* and *PT^R*) that promote the transition from a round shape to a pointed tip via altering of the affinity of this TF to downstream promoter genes. The authors also show that the *PT^R* locus was selected during the tomato domestication process, since wild tomato accessions do not produce fruit with pointed tips, while more than 21 % of the *S. lycopersicum* BIG accessions exhibit fruit with pointed tips. Using yeast one-hybrid, transient transactivation assay, EMSA and DNA-protein binding affinity assay, the authors reveal that PT can repress transcription of the *FUL2* gene by direct binding to its promoter. Moreover, they show that the PT/*FUL2* regulatory module controls auxin transport and distribution, therefore determining the development of round tomato or pointed tips.

Re: We thank the reviewer for the accurate summary of this study. Based on the reviewer's suggestions and comments, we have performed additional experiments and made careful corrections in the manuscript.

Although the manuscript does provide real novelty in identifying a fruit tip locus that has not been previously described, the study fails, however, to provide any histological information on how and when the fruit tip is initiated. More importantly, the manuscript contains some inconsistent data and a number of misinterpretations of the results. Also, the experimental design is not always optimal. Surprisingly, comparative global transcriptomic profiling is missing in this study whereas RNA-seq analysis could have provided insight into the gene networks underlying tip formation.

Re: We are grateful for the comments and suggestions. To pinpoint the anatomical stage when the fruit pointed tip is initiated, we dissected developing flowers and fruits at different developmental stages from CR-*pt^H* line and its wild-type TS-9. As compared with the wild-type TS-9, the ovary of CR-*pt^H* displayed no visible differences before and at anthesis. Subsequently, with the senescence of the style (wilted from the tip to the base), the distal end of the ovary from CR- *pt^H* gradually rose into fruit pointed tips, whereas the distal end of the TS-9 ovary stayed the same, indicating that the fruit pointed tip of CR-*pt^H* is initiated after pollination and fertilization and before the style abscission (Fig. 4a). Furthermore, we detached the style from the pollinated and fertilized flowers in the CR-*pt^H* line to explore the relationship between the presence of a style and the initiation of pointed tips. We found that the style-detached CR-*pt^H* line produced non-pointed tip fruit relative to the style-intact CR-*pt^H* fruit, which formed pointed tips. (Fig. 4b).

Additionally, we investigated the cell development of ovaries at different developmental stages using paraffin sections. We discovered no apparent abscission zone in the pointed tips, and the development of fruit pointed tip cells originated from the cell division and expansion of the CR-*pt^H* ovary (Fig. 4c). We added these data to the revised Fig. 4 and described the data in the revised manuscript on lines 245 to 268.

Regarding the inconsistent data and misinterpretations of our results, we have performed additional experiments and made careful corrections. On the other hand, as suggested, we conducted additional RNA-seq experiments in the distal end of the fruit from CR-*pt^H* and the wild-type control TS-9. GO enrichment analysis of differentially expressed genes (DEGs) showed that a large portion of enriched terms were found to be associated with the auxin polar transport and auxin homeostasis. We provided these data in the revised Supplementary Fig. 9 and Data 3.

The data related to the regulation of fruit ripening are confusing and weak. Actually, they are irrelevant to the scope of the paper and it would be rather beneficial to discard the part dealing with fruit ripening from the manuscript.

Re: As suggested, we deleted the data and descriptions related to the regulation of fruit ripening in the revised manuscript.

The pointed tip formation is an interesting topic when aiming to further understand the factors underlying the wide diversity of fruit morphology, but the agricultural importance of this trait has not been addressed in the manuscript which reduces the overall impact of the paper.

The manuscript suffers from several serious shortcomings that need to be thoroughly addressed.

Re: We appreciate the comments. Rough scars at the distal end of the fruit severely affect the quality of tomato fruit products. Selection of accession with pointed tip genotype is a common method used in breeding to overcome the occurrence of the rough blossom-end scarring in large-fruited, fresh-market tomatoes⁵. We have provided this information in the revised manuscript on lines 72 to 75 and 397 to 400.

Major comments:

1. A broad introduction of the formation fruit tip in tomato and in other fruit species is missing. What is known about the organ/tissue from which the fruit tips are formed and when this process starts to take place?

Re: The formation of fruit tips in tomatoes and other fruit species has, to our knowledge, not been studied extensively. A previous study has shown that overexpression of *SIFUL2* results in the formation of pointed tip fruits. The initiation of pointed tips has been attributed to the abnormal abscission of styles and ovary cell development during the pollination and fertilization stages⁶. We added this information in the revised manuscript on lines 84 to 87.

2. It is rather surprising that the GWAS didn't detect SNPs associated with other genes (those already known) involved in the pointed tip trait. What about also the detection of SNPs in the *FUL2* gene?

Re: GWAS is a valuable and effective method that can be used to reveal the genetic basis underlying

complex traits of different crops. However, our knowledge about the genetic basis of pointed tip fruit is very limited. Although pointed blossom-end fruit mutants have been identified, including *beaked* (*bk*), *bk-2*, *nipple-tip* (*n*), *n-2*, *n-3*, *n-4*, and *persistent style* (*pst*), none of them has been cloned. In our study, we identified several pointed tip fruit loci on chromosomes 2, 3 and 5, which are consistent with the *bk* and *n* loci reported in previous studies. Interestingly, we also detected an SNP in the 5'UTR of *FUL2* between the pointed tip and non-pointed tip fruit accessions. However, the physical position of the SNP was found to be more than 1 Mb away from the lead SNP on chromosome 3. This may be due to the fact that the SNP variation is less frequent in these 311 tomato accessions or is not the causal one underlying the pointed tip fruit. We added this information in the Discussion section of the revised manuscript on lines 456 to 469.

3. Is there a particular reason why accession LA4053 was chosen for the cross between round and pointed tip, as this netted skin line seems to have very strong and peculiar phenotypes?

Re: Yes, LA4053 presents a netted-cracking fruit phenotype. In addition to studying the pointed tip fruit, our group also used it to map and clone the genes underlying the netted-cracking fruit phenotype⁷. Multiple uses of one F₂ population can save a lot of human and financial resources.

4. It is not clear how the pointed tip trait is defined. This definition is lacking in both methods and legend to figures. When conducting the RT-PCR experiments and IAA quantification assays, there is no description on how the samples were collected from round and pointed-tip fruit?

Re: We apologize for the lack of definition for the pointed tip trait. Based on the angle of the distal end of the fruit, we classified the whole accessions into pointed tip fruits ($0^\circ < \theta < 180^\circ$, Fig. 1c) and non-pointed tips fruit, and we defined the protruding position of the distal end of the fruit as the pointed tip part. When conducting the RT-PCR experiments and IAA quantification assays, we collected the pointed tips part (pointed tip accessions) and corresponding pericarp part (non-pointed tip accessions) from the distal end of the fruit. We added this information in the Methods section of the revised manuscript on lines 530 to 533, 566 to 568, and 617 and revised Fig. 1c.

5. The expression at the transcript level is only provided in one accession, what about other genetic backgrounds? Also, at which stage of tip development *PT* is expressed? Expression analysis of *PT* gene at different stages spanning the tip formation period is missing.

Re: We have detected the *PT* transcripts in six accessions, including three pointed tip fruit accessions and three non-pointed tip accessions. The *PT* gene was found to be transcribed at similar levels in both groups of accessions. Additionally, the transcript levels of the *PT* gene at different stages spanning the pointed tip formation period were also quantified. We found that *PT* was transcribed during the early stages of fruit development, at pollination and fertilization, and before fruit enlargement, which is consistent with the progression of fruit tips development. We provided this information in the revised Supplementary Fig. 5 and revised manuscript on lines 193 to 199.

6. The expression patterns assessed by qRT-PCR and by GUS reporter fused to *PT* promoter do not seem

to match. What is the length of the promoter used to drive the GUS reporter gene? Does it include the 5'UTR?

Re: We used a 2,029-bp promoter to drive the expression of the *GUS* reporter gene, which was placed in-frame and upstream of the ATG codon of the *PT* coding sequence and contained the 5'-UTR. In the original manuscript, the qRT-PCR and GUS staining results did not entirely match. This may be caused by the low sensitivity of GUS staining. Therefore, based on the suggestions of Reviewer #5, we deleted the GUS staining data and replaced it with the quantified transcript level results, which included different stages spanning the pointed tip formation period.

7. Since *PT* is expressed in several tissues/organs, it is surprising that the KO mutation doesn't result in more phenotypes in addition to pointed tip. How many *PT* orthologs are in the tomato genome? Could it be that the absence of additional phenotypes is due to functional redundancy among homologs. A phylogenetic analysis of *PT* genes is missing. Also, a presentation of the C2H2 family of TFs and their function in the tomato and other plant species is missing in the introduction section.

Re: We appreciate the suggestions. In addition to the pointed tip fruit phenotype, *PT*-knockout mutant plants also exhibited curled leaves, thicker stems, and shorter styles. The molecular mechanisms underlying these phenotypes are subject of other currently ongoing studies and were not described and discussed in this manuscript. Additionally, a recent study has reported that transgenic plants with the *OBV* gene (namely *PT*) knocked-out by CRISPR (*Cris-OBV*) showed obscure leaf veins as compared with the wild-type¹. On the other hand, we conducted a phylogenetic analysis of *PT* paralogs and orthologs from selected plants and found that the tomato *PT* protein is located in an independent branch, indicating that *PT* may have distinct functions in the formation of pointed tip fruit in tomato. We provided this data in the revised Supplementary Fig. 2 and the revised manuscript on lines 179 to 182. As suggested, the information about the C2H2 family of TFs and their function was added in the Introduction section of the revised manuscript on lines 95 to 111.

8. It is surprising that *PT^R*-OE lines exhibit pointed tips since it is shown that the *PT^R* allele is recessive to *PT^H*.

Re: Thank you for pointing this out. Although this was a bit surprising, it might be reasonable. Regarding our finding that the *PT^R* overexpression lines produced fruit with pointed tips, we propose that *PT^R* might inhibit the activity of endogenous *PT^H* by forming heterodimers or tetramers with the endogenous *PT^H*. We discussed this possibility in the Discussion section of the former version manuscript.

9. The claim that *PT^R* promotes the formation of fruit with pointed tips and that *PT^H* allows development of round fruit is an overstatement. It seems that the pointed tip trait is dependent on the absence of the *PT^H* allele, regardless of whether or not the *PT^R* allele is present. This is supported by the phenotypes CR-*PT^H*-KO lines. Again, this makes highly questionable the presence of the pointed tip phenotype in the TS-9 lines expressing the *PT^R* transgene.

Re: Indeed, the pointed tip trait was dependent on the absence of the *PT^H* allele, and this was confirmed by

the fruit phenotypes in the CR- PT^H lines. Moreover, PT^R allele appeared to be a loss of function for fruit pointed tip according to the phenotype of the F2:3 harboring either PT^H and PT^R alleles plants and CR- $pt^{H/R}$ knockout lines. We deleted this overstatement conclusion in the revised manuscript.

10. It is rather confusing that the authors did not use the same genetic background of round and pointed tip accessions for the generation of the different mutants. Using the same genetic background all over the study would have facilitated the comparison of the impact of all mutations (overexpression, knockout). When trying to decipher the functional significance of PT, going from the LA4053 accession, to TS-3 and then to Micro-Tom for the pointed-tip lines makes the interpretation of the data more difficult as they all display variable traits in addition to the pointed tip.

Re: We appreciate for the comments. In this study, different accessions were used for different research purposes. TS-9 and TS-3 were used for functional complementation experiments in GWAS analysis. LA4053 was used for functional complementation experiments in F₂ populations. Nevertheless, we used the same genetic background (TS-9) for all the genetic and molecular mechanism research experiments. We deleted the Micro-Tom accession in the NPA treatments experiment and replaced it with the parental accession and PT^H -KO mutants (revised Fig. 6).

11. Figure 3B shows clear pointed tips in PT^R -OE lines, but the transcript level of *FUL2* (Fig. 4A) was not altered. This is inconsistent with the pointed-tip being dependent on high expression of *FUL2*. In my opinion, OE- PT^R lines are not the good control to use for evaluating the function of PT^R and for addressing its physiological significance in comparison to that of its PT^H allelic variant. Instead of overexpressing 35S- $PT^{H/R}$ in PT^H genetic background, it would be more informative to express proPT: $PT^{H/R}$ in the PT^H -KO or PT^R -KO lines.

Re: Thank you for the suggestion. PT^R allele is a loss of function for fruit pointed tip. As pointed out by this reviewer, OE- PT^R lines are not a good control to use for evaluating the function of PT^R and for addressing its physiological significance in comparison with that of its PT^H allelic variant. Therefore, following your suggestion, we expressed proPT: PT^R in the CR- pt^H lines and found that the CR- $pt^H/PT^R_{pro}:PT^R$ plants produced fruit with pointed tips. We added this data in the revised Fig. 3l and Supplementary Fig. 6g and 7g.

12. Figure 4E, 4F and 4G are not convincing in showing the binding specificity of PT^H compared to PT^R .

Re: In our study, Figure 4E, 4F and 4G (revised Figure 5e, 5f and 5g) show that PT^H binds to the *FUL2* promoter with higher affinity than PT^R . Therefore, we drew the conclusion that PT^H directly binds the *FUL2* promoter with higher affinity than PT^R . We did not state that PT^R cannot bind to the *FUL2* promoter. Additionally, we also added the data of yeast cells co-transformed with pAbAi-*FUL2* and pGADT7- $PT^{H/R}$ growing on the SD/-Ura-Leu selective medium with lower AbA concentrations and found that yeast cells co-transformed with pAbAi-*FUL2* and pGADT7- PT^R could grow weakly on the SD/-Ura-Leu selective medium containing 40 ng ml⁻¹ AbA, which was consistent with the results in Figure 4E, 4F and 4G (revised Fig. 5e, 5f and 5g). We provided this information in revised Fig. 5c.

13. In order to demonstrate the epistatic relationships between PT and FUL2 genes, I am not sure that the most appropriate genetic experiments are those reported (Figure 4J). The CR-ful2/CR-pt^H double mutant is not meaningful in this regard. More convincing would be to express the 35S::FUL2 gene in the 35S::PT^H genetic background as in this way FUL2 should escape the PT^H-dependent down-regulation. Therefore, if PT^H is really acting via down-regulation of FUL2, then these lines should display the pointed tip phenotype despite the presence of a functionally active PT^H protein.

Re: In this work, CR-pt^H mutants displayed pointed tip fruit phenotype, and FUL2 was upregulated in CR-pt^H mutants relative to the wild-type TS-9. In order to characterize the genetic relationships between PT and FUL2, we crossed the ful2 mutants with CR-pt^H mutants. We found that CR-ful2/CR-pt^H double mutants present non-pointed tip fruit and FUL2 could escape the CR-pt^H-dependent up-regulation. Therefore, we conclude that CR-ful2 is epistatic to CR-pt^H. Additionally, the development of oval-shaped fruit and necrosis at the distal end of the fruit from PT^H-overexpressing lines were likely due to the strong expression of the 35S promoter rather than the PT^H allele itself. Thus, we believe that overexpression might not be an ideal approach for demonstration of their epistatic relationship and instead, application of double mutants might be more suitable for revealing the genetic relationships between two genes.

14. In Figure 5A and 5B, there is no description on how the NPA and PCPA treatments have been performed. Did the authors perform injection of the flower or the fruit? At which stage? How many times?

Re: We apologize for missing the NPA and PCPA treatment information. For this experiment, flowers of anthesis from TS-9, LA4053, and CR-pt^{H/R} plants were sprayed with 100 mg/L NPA or 20 mg/L PCPA, respectively. The treatments were alternated every day and each type of the hormones were sprayed three times. We provided this information in the legend of revised Fig. 6.

15. The DR5:GUS expression data presented in Figure 5C are not convincing. It is not obvious that the blue staining is more intense in CR-pt^H than in TS-9. This is not surprising since, to my understanding, the crossing gives heterozygous lines in which the PT^H gene is still able to produce enough amount of the corresponding protein to repress the expression of FUL2 gene.

Re: We thank the reviewer for pointing this out. Indeed, the DR5:GUS expression data were not convincing. The GUS expression was weak, diffuse, and not specific to the tip. Thus, following the suggestions of Reviewer #5, we deleted the DR5:GUS expression data.

16. Figure 5A and 5B show that tip formation might be associated with auxin transport, but they are using different genetic background. A more solid way would be to perform NPA and PCPA treatment directly in PT^{H/R} OE and KO lines.

Re: PT^H and PT^R KO lines and their wild-type controls have been used for the NPA and PCPA treatment in the revised manuscript, accordingly. Regarding the PT^{H/R} OE lines, we think that they may not be suitable for NPA and PCPA treatments because the overexpression of PT alleles under the 35S promoter may have collateral effects. We added this data in revised Figs. 6a-e.

17. In all figures showing fruit pictures, there is no indication on how many fruits are being tested and what is the percentage of fruits showing round or pear shape with/without pointed-tips. Some figures lack statistical analysis. For example, Fig.4E and Fig. 5A-C. In Fig 5C, it is difficult to conclude on whether the spatial expression patterns of DR5-GUS are totally different among the three lines or whether they only differ with regard to the expression in the distal part of the fruit. In most figures, it is not stated how many fruits from each line were used to conduct the experiments?

Re: In this study, three replicates per line and 20 fruits per replicate were used for most of the experiments. We calculated the percentage of fruit showing pear or pointed tip and provided the statistical analysis data in revised Supplementary Figs. 7, 12 and Fig. 6. Because of the weak and diffused *GUS* expression patterns, we have deleted the *DR5:GUS* expression data from the revised manuscript.

18. The early ripening story is difficult to understand because it defies the principles of what is widely accepted regarding the mechanisms of climacteric fruit ripening. The high expression of *ACO1* at 24 dpa (Figure 6A) can hardly be correlated with the early ripening. At this early immature green stage, the fruit didn't gain their competence to ripen. Moreover, the data in Figure 6A and Figure 6B are not consistent and *ACO1* expression unlikely reflects the early ripening. Indeed, it is shown in the Figure that at 34 dpa *ACO1* expression is similar in WT and PT^R-OE fruit while at the same stage (34 dpa) ethylene production is 5 times higher in PT^R-OE. Does similar enhanced ethylene production occur also in CR-pt^H fruit? Surprisingly, Figure3L seems to indicate that pt^H-KO lines don't display early ripening while they are anticipated to produce more FUL2 proteins like PT^R-OE lines. Do the tomato accessions bearing the *PT^R* allele show faster fruit ripening than those with *PT^H* allele?

Re: We sincerely thank the reviewer for raising these questions. Indeed, early ripening, *ACO1* expression levels, and ethylene production were not fully consistent in this study. As commented by this reviewer, the regulation of fruit ripening is irrelevant to the scope of the manuscript. We have deleted this part of data and discussion from the revised manuscript.

19. Figure 6F shows that PT^R but not PT^H is able to bind the DNA fragment containing the A(G/C)T motif, while Figure 4E shows that PT^H binds the A(G/C)T containing fragment with higher affinity than PT^R. This is fully inconsistent. Keeping with the same idea, Y1H shows that PT^R doesn't bind to FUL2 promoter (Figure 4C), but this is inconsistent with the outcome of the EMSA experiments showing that PT^R binds to FUL2 promoter fragment (Figure 4E).

Re: In this study, we identified an amino acid variant in the highly conserved C2H2 domain of PT proteins. The change in the number of C2H2 domain may affect the affinity of the PT protein for particular DNA sequences and therefore, alter the ability of PT to regulate the transcription of particular genes. This ability to bind may also be related to residues at the boundaries of the conserved element. We are sorry for drawing the inaccurate and inconsistent conclusion. We revised this conclusion, which now reads "These results indicate that PT^H directly binds the *FUL2* promoter fragments with higher affinity than PT^R". Regarding the inconsistency of Y1H and EMSA experiments in Figure 4 (revised Fig. 5), we believe that it

the two systems have different sensitivities, and the results are not contradictory. Our conclusion from the EMSA experiment is that PT^H directly binds the *FUL2* promoter fragments with higher affinity than PT^R , which does not exclude the possibility of PT^R to bind to the *FUL2* promoter (Fig. 5e). We added the data of yeast cells co-transformed with pAbAi-*FUL2* and pGADT7- $PT^{H/R}$ growing on the SD/-Ura-Leu selective medium with lower AbA concentrations and found that yeast cells co-transformed with pAbAi-*FUL2* and pGADT7- PT^R could grow weakly on the SD/-Ura-Leu selective medium containing 40 ng ml⁻¹ AbA, which is consistent with the conclusion from the EMSA experiments. We provided this result in revised Fig. 5c.

Additional comments:

1. It is not true that all fruit develop from the ovary (Line 62). Strawberries for example develop from the receptacle portion of the stem of the flower.

Re: The statement was corrected and now reads "Most fruits are specialized organs that develop from ovaries after successful pollination and fertilization".

2. How to define the pointed tip trait? To which degree of the angle formed at the distal end of tomato fruit it is called pointed-tip? This definition is lacking in both methods and figures. Round fruit and pointed-tip are two different traits. Although most round fruits are shown without tips, I think the author could change "round fruit" to "no tips" as this is how they conducted the GWAS study.

Re: We apologize for the lack of definition of 'pointed tip' trait in the original manuscript. Based on the angle of the distal end of the fruit, we classified the whole accessions into pointed tip fruits ($0^\circ < \theta < 180^\circ$) and non-pointed tip fruits. We defined the protruding position of the distal end of the fruit as the pointed tip part. We provided this definition in revised Fig. 1c and in the revised manuscript on lines 530 to 533. Additionally, we have changed "round fruit" to "non-pointed tip fruit" in the revised manuscript.

3. In Methods section, when conducting the RT-PCR experiments and IAA quantification assay, how were the pointed-tips collected from fruits, especially those materials giving round fruit. It is not clearly explained in the manuscript.

Re: When conducting the RT-PCR experiments and IAA quantification assays, we collected the pointed tip part from the pointed tip accessions and the corresponding pericarp part from the non-pointed tip accessions from the distal end of the fruit. We added this information in the Methods section of the revised manuscript on lines 530 to 533, 566 to 568, and 616 to 617.

4. The title of Figure 2 is not correct." Loss of one C2H2 domain in PT^R from an adenine (A) to guanine (G) mutation in *PT*". Similarly, the title of Figure 3 is confusing "Loss-of-mutations in both PT^R and PT^H alleles that led to the formation of fruit with pointed tips". It is not clear what loss-of-mutation means? The titles of most figures are not well written and need editing.

Re: The title of Figure 2 was changed to "The G/A polymorphism in *PT* leads to the change of C2H2 domain number". The Figure 3 title was changed to "Mutation of *PT* led to the formation of fruit with

pointed tips". Additionally, we checked all the figure titles and revised them, accordingly.

5. The CR lines generated for *PT*, *FUL2* and *ACO1* genes in the three tomato accessions are not well represented in Fig. 3I-K, Fig. 4I and Fig. 6I. Are they all KO lines? Which regions are represented for the C2H2 or other conserved domains? The predicted protein from CDS is not indicated in the figures.

Re: We are sorry for the lack of the edited genomic sequences of the CR lines. The CR lines used in this study were all KO lines. Except CR-*pt^H*-4 line, all other CR-lines contained either an insertion or a deletion in the target region of the *PT* gene that would change the reading frame, thus generating knockout mutant lines. For CR-*pt^H*-4 line, the predicted PT protein would contain deletion of three amino acid residues in the conserved spacers between zinc fingers. We provided the predicted amino acid sequence information in revised Supplementary Figs. 8 and 11.

6. Does *PT* expression at the protein level is similar in round and pointed tip mutant? It would be more correct to say that the expression is assessed at the transcript level only.

Re: We have corrected it and the statement now reads "We found that *PT* was transcribed at similar levels in the pointed tip and non-pointed tip fruit accessions" in the revised manuscript on line 193.

Reviewer #3 (Remarks to the Author):

Fruit shape defined by several features of the fruits is an important aspect of fruit quality and consumer acceptance by and large. Some tomato varieties, especially cherry tomato, produce pointed fruits. Though auxin over-production has been linked to outgrowth of the distal end of the fruit, causing the formation of pointed fruits, the genetic basis of this trait is not known. Song et al in this paper report the identification of natural variants associated with the formation of pointed fruits in tomato. Using 311 tomato accessions previously re-sequenced, they by GWAS analysis identified a SNP in the coding region of Solyc05g054030 highly associated with the pointedness of the distal fruits. An elegant phenotypic analysis of loss-of-function mutants and overexpression lines of Solyc05g054030 in different genetic backgrounds confirmed the H135R mutation is response for the formation of the pointed tips. The authors hypothesized that the MADS gene *FUL2* is a potential *PT* target based on a similar phenotype they observed on *FUL* overexpression lines instead of performing a genome-wide identification of *PT* targets. Fortunately, the authors have been able to show that the zinc finger protein *PT* binds to *FUL* promoter by several in vitro assays and in *N. benthamiana* leaves, further supported by a genetic analysis using *pt ful* double mutant. As previous work has shown that pointed tips often happen in the fruits over-producing auxin, the authors confirm the formation of the pointed tips controlled by *PT* is also attributed to high auxin signals in the distal ends of the fruits. Overall, this is an interesting story, though the significance of these findings is arguable.

Re: We thank the reviewer for the summary of this study. Rough scars at the distal end of fruit severely affect the quality of tomato. Selection of germplasms with pointed tips is a common method used in breeding to overcome the rough blossom-end scarring in large-fruited, fresh-market tomatoes⁵. We added

the information of the significance of our findings in the revised manuscript on lines 72 to 75 and 397 to 400.

Despite PT is within a domestication sweep and the *PT* (R) allele is prevailing in large fruit accessions, it is debatable that the trait was targeted during domestication. Pointed fruits apparently have no contribution to the adaptation to growth conditions or nutrient improvement in tomato. More likely the H135R mutation was rose and somehow retained in cultivated tomato, unless there are unknown traits regulated by PT that was targeted by domestication.

Re: We appreciate for the comments. In a recently published report¹, the *PT* gene was identified as the "*OBSCURE VEIN*" (*OBV*) and their study has suggested that the *obv* mutation may have hitchhiked the selection for mutations in the nearby *SP5G* (*SELF PRUNING 5G*), which leads to more compact plant growth habits and shorter growing season^{1,2,3}. We added this "hitchhiked" hypothesis about the possibility of *PT* and other nearby loci as targets of selection during domestication in the Discussion section of the revised manuscript on lines 480 to 492.

Solyc05g054030/PT from the sequenced reference genome of Heinz1706 also contains the R135 variant like those from TS-3 and LA4053, but Heinz1706 set fruits without pointed tips. This suggests that the R135 variant in this gene is not solely response for the phenotype.

Re: Indeed, the *PT* sequence of the reference genome of Heinz 1706 also encodes the R135 variant. Heinz 1706 does set fruits with prominent pointed tips, which is consistent with the conclusion of this work. In this study, the accession TS-253 is Heinz 1706 and its pointed tip phenotype of fruit is shown in Fig. 1d.

Solyc05g054030/PT shares high sequence similarity with *Arabidopsis* IDD15/SGR5 that together with IDD14 and IDD16 regulates auxin biosynthesis and transport and is involved in organ morphogenesis and gravitropic responses in *Arabidopsis*. In tomato, it may have different functions, but these INDETERMINATE (ID)-DOMAIN proteins (IDDs) have four zinc finger domains (ZF1-ZF4) (Colasanti J, Tremblay R, Wong AY, Coneva V, Kozaki A, Mable BK. The maize INDETERMINATE1 flowering time regulator defines a highly conserved zinc finger protein family in higher plants. *BMC Genomics*. 2006 7:158). Does PT truly contain two ZF domains? Discussion on the functional diversification/divergence between PT and IDDs from other species is warranted.

Re: Thank you for the comments and suggestion. The most homologous genes of *PT* in *Arabidopsis* are *IDD15*, *IDD16*, and *IDD14* (Supplementary Fig. 2), which possess 4 zinc finger (ZF) domains. In comparison, tomato *PT*^R and *PT*^H contain only 2 and 3 ZF domains, respectively (Fig. 2a). In this study, the SMART database (<http://smart.embl-heidelberg.de/>) was used to predict the conserved domains of PT proteins. Phylogenetic analysis shows that PT protein is placed in a separate branch, indicating that PT may have distinct functions in the formation of fruit in tomato (Supplementary Fig. 2). Interestingly, tomato *SE3.1* (Solyc03g098070), the most homologous gene of *Arabidopsis* *IDD15* (Supplementary Fig. 2), also contains only 3 ZF domains and is involved in the style extension and self-fertilization⁸. Therefore, the IDD transcription factors may present sequence and functional diversification in tomato and

Arabidopsis. We provided this discussion in the revised manuscript on lines 419 to 432.

FUL2 (and *FUL1*) has been shown to regulate fruit ripening in tomato (Bemer M et al., 2012 Plant Cell 24(11):4437-51; Fujisawa M et al., 2014 Plant Cell 26(1):89-101). If PT targets *FUL2* as shown in Fig 4, early ripening observed in fruits overexpressing the PT (R) allele may be caused by disrupted *FUL2* activity. The possibility may need further investigation.

Re: Thank you for the comments. *FUL1* and *FUL2* have a redundant function in fruit ripening and their double silenced lines exhibit an orange-ripe fruit phenotype due to highly reduced lycopene levels⁹. In our study, PT^H was shown to target the expression of *FUL2* to underlie the non-pointed tip fruit phenotype, whereas PT^R targets *ACO1* gene expression to regulate fruit ripening. Because fruit ripening is irrelevant to the scope of the manuscript, we have deleted the fruit ripening data and description in the revised manuscript.

Reviewer #4 (Remarks to the Author):

The authors describe how they by a combination of GWAS and map-based cloning, identified a single SNP is the coding region of a C2H2 Zinc finger transcription factor that is responsible for the presence/absence of a pointed tip on tomato fruit. The "wild" allele appears to code for three Zn fingers, while the derived allele contains a disruption in one of the three, apparently causing the pointed tip. They then show that the full 3 Zinc finger version likely suppresses *FUL2* expression by direct interaction with the promoter and that interference with that process causes a local increase in auxin, in turn causing the tip phenotype. Conversely, the derived allele encodes a protein not able to suppress *FUL2* but in turn activating *ACO1*, ethylene production, and early ripening (Fig. 8). The molecular characterization looks good but contains a few gaps that I would like to see resolved (see below).

Re: We thank the reviewer for the summary of this study. The revised manuscript contains corrections throughout the text and data of additional experiments, and we hope that the gaps are filled.

Furthermore, as good as the molecular characterization may be, a good developmental basis for the phenotype is lacking, which is a pity. What particular developmental process causes the tip phenotype? Is it determined already in the ovary? Or during what stage? What does the tip consist of? Local increase in cell division or expansion? Only a single figure (1J) is shown without further elaboration. Furthermore the phenotyping for "pointed tip" lacks a quantitative description. When is a tip considered pointed? In Fig. 1C, the difference between TS-9 and most of the others seems obvious, but TS-246? Most pointed fruits are also elongated. Is there a link? Fig. 1K seems to indicate that there at least two more loci involved.

Re: We are grateful for the critical comments. To pinpoint the stage of the initiation of fruit pointed tips, we dissected the developing flowers and fruits at different developmental stages from the CR-*pt^H* line and its wild-type TS-9. As compared with that of the wild-type TS-9, the ovary of CR-*pt^H* displayed no visible differences before and at anthesis. Subsequently, with the senescence of the style (wilted from the tip to the base), the distal end of the ovary from CR- *pt^H* gradually extended to form the fruit pointed tips. In

comparison, the distal end of the TS-9 ovary did not grow as the style senesced, indicating that the formation of the fruit pointed tips in CR-*pt^H* is initiated after pollination and fertilization and before the abscission of styles (revised Fig. 4a).

Additionally, we investigated the cell development of ovaries at different developmental stages using paraffin sections. We discovered no apparent abscission zone in the pointed tips, and the development of fruit pointed tip cells originated from the cell division and expansion of the CR-*pt^H* ovary (revised Fig. 4c). We added these data to the revised Fig. 4 and described the data in the revised manuscript on lines 245 to 268.

Regarding the quantitative description of "pointed tips", we defined pointed tip fruits as having the angle (θ) of the distal end of the fruit in the range of $0^\circ < \theta < 180^\circ$. We used this definition to classify the whole tomato accessions into two groups, pointed tip fruit and non-pointed tips fruit. We provided this definition in revised Figs 1c-1d and the Methods section of the revised manuscript on lines 530 to 533.

As to the possible correlation between pointed tip fruits and elongated fruits, our observation showed that only some of the pointed tip fruits, rather than most of them, are also elongated. In particular, CR-*pt^H* of this study developed round fruits with pointed tips (revised Figs. 3i, 1), suggesting that the pointed tip trait and the elongated fruit phenotype are not linked. Therefore, to avoid misunderstanding of the possible correlation between the two fruit traits, we added images of some elongated fruits without pointed tips in revised Fig. 1d. Indeed, based on the GWAS and BSA analyses, we identified several loci underlying the pointed tip trait. We discussed these loci in the Discussion section of the revised manuscript on lines 456 to 469.

In their discussion, the authors suggest that pointed tips may have been selected for during domestication due to their consumer appeal? I find that hard to believe & did the authors check whether the pointed tip accessions are really consumer tomatoes rather than processing ones? The authors are forgiven for not noticing that last year, probably during the preparation of their manuscript, another group identified exactly the same SNP as underlying the "obscure vein" (*obv*) mutation (Lu, J. et al. (2021). *OBV* (obscure vein), a C2H2 zinc finger transcription factor, positively regulates chloroplast development and bundle sheath extension formation in tomato (*Solanum lycopersicum*) leaf veins. *Hortic. Res.* 8.). Just as those authors do not mention the pointed tip phenotype, in this study surely the *obv* phenotype must have been apparent? It would be good to discuss this. It was suggested that the *obv* mutation has "hitchhiked" along with selection for mutations in the nearby *SP5G* which leads to more compact plant habit and shorter growing season. For PT, the same may very well apply, more likely than consumer preference.

Re: We appreciate the comments and suggestions. We are sorry to miss the reference of Lu et al., 2021, which was indeed published while the first version of this manuscript was prepared. In that report, they have identified the same SNP underlying the "*obscure vein*" (*obv*) mutation as the one in *PT^{H/R}* in our work. Their observation reveals that overexpression of *OBV* (i.e. *PT^H*) results in a change from the dark veins of leaves into a transparent vein phenotype¹. Consistent with their observations, we also noticed that *PT^H*-overexpressing plants also developed transparent veins as compared with the dark veins in the wild-type TS-9 control. Together with two other studies on the *obv* mutation, these authors propose that the

obv mutation may have hitchhiked the selection for mutations in the nearby gene *SP5G* (*SELF PRUNING 5G*), which leads to development of more compact plant growth habits and shorter growing season^{1,2,3}, which apparently offers more beneficial effects on tomato growth and production and is thus more likely to be a target of selection during tomato domestication. We added this information in the Discussion section of the revised manuscript on lines 482 to 492.

Specific comments:

Title: is not cover the contents very well

Re: It was corrected to "Variation in the fruit development gene *POINTED TIP* results in protuberance at the stylar-end of tomato fruit".

line 78: what is "fruit flatness" Did you mean surface smoothness (as opposed to ribbed or blocky)?

Re: In this study, fruit flatness meant the flat degree of fruit shape, but not the surface smoothness. We changed the term with "locule number", which is positively correlated with the flat degree of fruit shape, and has been found to be controlled by the *locule number* (*lc*) and *fasciated* (*fas*) loci' in the revised manuscript on lines 50 to 51.

line 143: "raised" requires a quantitative measure

Re: We have revised the word "raised" with "the angle of the distal end of the fruit ($0^\circ < \theta < 180^\circ$)" and provided this information in revised Fig. 1c and the manuscript on lines 133 to 135.

line 203-204: *PT* is highly expressed in leaves, flowers, and styles, but much lower in ovary and immature fruit; this is also relevant for the question where and when *PT* action takes place. The GUS activity of the promoter fusion (fig. S5B is not very clear, but seems to show that expression is particularly high in the distal end of the ovary. Expression of *PT* could have been determined for the distal part of the immature fruit to see whether it is locally high there vs. other parts of the fruit. Overexpression of *FUL2* causes delayed style abscission (Wang et al). Is that the case in *PT(R)* fruits as well. What is the relation with the pointed tip? See also line 257

Re: Thank you for the comments. To explore when and where the *PT* action takes place during flower and fruit development, we performed anatomical observations of the developing pointed tips at different developmental stages of floral and fruit formation in *CR-pt^H* plants in comparison with the wild-type TS-9 control (also see reply above). The GUS activity of the promoter fusion was relatively weak and diffused. Therefore, based on the suggestion of Reviewer #5, we have deleted the GUS staining data and replaced it with quantified expression results, which included different stages spanning the pointed tip formation period (revised supplementary Fig. 5). To investigate the relationship between the style abscission and pointed tip formation, we detached the style from the pollinated and fertilized flowers in the *CR-pt^H* lines. We found that the style-detached *CR-pt^H* plants produced non-pointed tip fruit as compared to the style-preserved *CR-pt^H* plants, indicating that style senescence is required for the formation of the fruit pointed tips. We added this data in revised Fig. 4b and described it in the revised manuscript on lines 256

to 261.

Line 312: Why Micro-Tom? in my experience MT has only pointed tips some of the time (mostly correlated with parthenocarpic development), not always. Does it contain the pt SNP?

Re: It contains the *PT* SNP. We have replaced it with the parents of the mapping population. We provided this data in revised Figs. 6a-e.

Line 348 and onwards: PT(R) overexpression is not a good proxy for determining its function. This should be observed in isogenic lines with either of the two alleles and confirmed by knockout in the PT(R) lines.

Re: We agree with the comment. For pointed tip fruit phenotype, we have observed in F2:3 population lines with either of the two alleles and confirmed by knockout in the $PT^{H/R}$ lines. However, based on the suggestions of Reviewer #2, the data and description about the regulation of fruit ripening has been deleted in the revised manuscript.

Lin 458-460 seems to be wrong

Re: We are sorry for the careless expression. It was corrected to " PT^H is a transcriptional repressor that attenuates *FUL2* expression by binding to the *FUL2* promoter with higher binding affinity than PT^R , which inhibits the accumulation of auxin at the distal end of the fruit and the development of fruit with pointed tips" in the revised manuscript on line 458.

Reviewer #5 (Remarks to the Author):

The manuscript entitled "Two natural alleles of *POINTED TIP* regulate distinct processes in tomato fruit" describes the mapping and cloning of *PT* by GWAS, biparental mapping and plant transformations. The two alleles are reported to differ in protein sequence resulting in a C2H2 protein with the predicted 2 or 3 zinc finger domains. The difference in two versus three domains leads to a dramatic difference in efficiency of promoter binding to either *FUL2* or *ACO1*, respectively, in a Y1H and EMSA. One allele PT^R , the zinc finger appears to function as a transcriptional activator whereas for the other allele, PT^H , the protein appears to function as a transcriptional repressor as measured in tobacco cells or leaves. The authors show a link to auxin transport for PT^H , leading to round fruits. The allele producing pointed fruits, PT^R , is effectively a loss-of-function allele of PT^H with respect to bottom fruit shape. Loss of function of PT^H also leads to earlier ripening (see figure 3L), suggesting that this locus controls ripening as well. The role of PT^R if any, is less clear in the plant. Whereas overexpression of PT^R leads to earlier ripening, the converse, a loss of function of PT^R does not lead to delayed ripening (fig 3M). This leads this reviewer to assume that PT^R in the plant is likely a loss of function for both bottom fruit shape as well as ripening, even though molecular studies such as promoter binding in gel shifts and Y1H support a role in transcriptional activation.

Re: We appreciate the summary of this work. Based on the existing data, we agree with the conclusion that PT^R in the plant is most likely a loss of function for both bottom fruit shape and ripening (see also reply

below).

Regardless, the study is quite interesting, and many parts of the research are solid with additional evidence to support the interpretation of the findings (figs 1-2). Other parts could be improved by additional experimentation to derive at the most plausible explanation. Fig 3B shows that overexpression of PT^R overrides wild type function of PT^R presumably by creating a dominant negative complex or a poisoned interaction of the two alleles. This could explain the observed pointed shape and early ripening in 3B which is the most likely interpretation when taking the whole of figure 3 into account. Yet, the authors do not consider this as a reasonable explanation of the phenotype observed in Fig 3 and instead assume that PT^R plays an active role in regulating ripening. As mentioned above, if PT^R was actively regulating ripening, the KO of this allele (Fig. 3M) would have resulted in delayed ripening. The interpretation of phenotypes observed by overexpression should always be taken with caution. The creation of a pear or oval shaped fruit was only found in lines overexpressing the PT^H allele, further supporting that this allele and not PT^R is the only functional allele. Therefore, the downstream molecular experiments, specifically the *ACO1* promoter analyses, are meaningless if this gene is not affected by expression of PT^R in vivo. This situation can easily be clarified if the authors were to conduct RNA seq with the near isogenic lines (from Fig 3A but ideally more inbred than an F2:F3), as well as with the CR and OX of PT^H and PT^R lines. Moreover, RNA seq would also need to be conducted in order to validate the notion that "only" auxin and "only" ethylene regulate pointed tip and ripening respectively.

Re: We greatly appreciate the reviewer's suggestions. Regarding the role of *PT* in the formation of pointed tips, we agree with the conclusion by the reviewer that the fruit tips presented by overexpression of PT^R might be caused by creating a dominant negative complex or a poisoned interaction of the two alleles. We discussed this possibility in the Discussion section. However, this speculation on the role of PT^R would need further studies in future. Instead of using the phenotype of PT^R -overexpressing lines, we expressed $PT^R_{pro}:PT^R$ in CR- pt^H lines, and found that the CR- $pt^H/PT^R_{pro}:PT^R$ plants produced fruit with pointed tips. We added this data in revised Fig. 3I. Regarding the role of *PT* in fruit ripening, we only recorded the fruit development time (DPA) of PT^R -overexpressing lines and wild-type control, which was not sufficient for drawing meaningful conclusions. As commented by the reviewer, the early ripening trait should be observed in near-isogenic lines with either of the two alleles and confirmed by knockouts in the CR- PT^R lines. Based on the existing data, PT^R acts most likely as a loss of function allele in plants for both the bottom fruit shape and ripening time. As suggested by Reviewer #2, we have deleted the data and discussions about the role of fruit ripening from the revised manuscript, as the regulation of fruit ripening is beyond the scope of this manuscript.

To confirm the role of auxin in the formation of fruit pointed tips, we conducted an RNA-seq analysis on the pointed tip tissue of CR- pt^H and the wild-type TS-9. GO enrichment analysis of differentially expressed genes (DEGs) from the RNA-seq experiment showed that a large portion of enriched terms were found to be associated with the processes of auxin polar transport and auxin homeostasis, indicating that the PT^H allele may exert its role in the initiation and formation of fruit pointed tips by regulating auxin transport and auxin homeostasis during fruit development in tomato. We added this data in revised supplementary

Fig. 9 and Data 3.

The last part of the study regarding whether PT^R was selected during domestication is weak. If the PT^R was selected during domestication, the allele should be fixed in modern (aka BIG) tomato and it is not. I would remove the last figure and instead only focus on the most interesting evolutionary aspect of the phenotype is that many pointed shaped tomato varieties carry this PT^R allele.

Re: In this study, we calculated the nucleotide diversity (π) between groups. We found that the ratios of nucleotide diversity between PIM and CER accessions were higher than the threshold, whereas the ratios of nucleotide diversity between CER and BIG accessions were lower than the threshold, indicating that PT was potentially selected during tomato domestication (Figs. 7a, b). Additionally, PT is part of a previously reported domestication sweep¹⁰. The frequency of the PT^H and PT^R alleles in the accessions from the wild tomato group was 98.2% and 1.8%, respectively. The frequency of PT^R gradually increased in the CER (8.6%) and BIG (24.5%) groups (Fig. 7c), suggesting PT^R could be a target of selection during domestication. However, as pointed out by Reviewer #1, the results of Figure 7 could be interpreted differently, because the pointed tip trait offers no significant benefits in plant growth, fruit yield and adaptation. Instead, the PT^R allele might have been preserved as a result of hitchhiking of this allele along with other locus or loci in the proximity that is/are the real target of selection during domestication. This interpretation is consistent with the observation of a recent paper by Lu et al., 2021, in which, the SNP underlying the "obscure vein" (*obv*) mutation is the same as the PT^R allele in our work. It has been suggested that the *obv* mutation may have hitchhiked the selection for mutations in the nearby *SP5G* which offers apparent growth and adaptation advantages^{1,2,3}. Therefore, we have adopted the hypothesis that the PT^R allele, namely *obv*, may have hitchhiked the selection for mutations of the nearby *SP5G*, which contributes to higher carbon assimilation rate and higher fruit yield in tomato. Thus, we have toned down the claim of PT^R as a target locus of selection in the revised manuscript. Instead, we have focused more on the distribution of the pointed tip tomato varieties that carry the PT^R allele in the revised Results section (lines 373 and 389), and discussed this information in the Discussion section of the revised manuscript on lines 482 to 492.

Lastly, the introduction nicely introduces the known loci controlling pointed shape in tomato, such as *bk*, *bk2*, *n* and *pst*, the discussion never relates these known mutations to the PT locus. While these old mutants have been around forever, the underlying genes have not been identified. Therefore, this study is potentially clarifying the molecular underpinnings of mutations that have been described 100 yrs ago. Thus, is PT corresponding to *n*? nipple tip? It looks like that *bk* and *pst* correspond to the other loci mapped in figure 2. The authors should clarify which of the known loci PT corresponds to.

Re: We appreciate the comments and suggestions. Based on the known information of the approximate map positions of the mutants on chromosomes, PT is most likely corresponding to *nipple tip*, whereas *bk* and *pst* may correspond to the loci mapped on chromosomes 2 and 7 in GWAS analysis, respectively. However, there is only limited available information on these legendary loci and more in-depth investigations using the pertinent mutants are warranted in future studies. Additionally, our GWAS analysis

also revealed an SNP in the 5'-UTR of *FUL2* between the pointed tip and non-pointed tip fruit accessions. However, the physical position of the SNP was found to be more than 1 Mb away from the lead SNP on chromosome 3. This may be due to the fact that the SNP variation is less frequent in these 311 tomato accessions or the *FUL2* locus does not lead to the formation of pointed tip fruit. We provided this information in the Discussion section of revised manuscript on lines 456 to 469.

Specific comments are below:

- In line 128, the authors refer to the "nucleotide substitution from G to A" is incorrect. A is wildtype and therefore the substitution is A to G.

Re: It was corrected as suggested. Thanks.

- Line 146, replace high for many.

Re: It was corrected as suggested.

- Line 148-149. The minor allele frequency (MAF) is a frequency. If 5 out of 311 is the cut off, that is a MAF of $5/311 = 0.016$. Please reanalyze the data with the correct MAF of 0.05 (which is the best MAF) which is 16 or more.

Re: In our previous manuscript, we analyzed the data with the MAFs ≥ 0.05 , the number of varieties with minor alleles was ≥ 16 . We modified it to "16" in the revised manuscript on line 139.

- Make sure that if referring to recessive nature of the trait (pointed tip) the authors back that up with showing the number of plants with tip and number of plants with round fruit (and intermediate if they were observed as well). To confirm the recessive nature of PT^R , please present the number of plants that are homozygous PT^H , homozygous PT^R and heterozygous with the phenotype.

Re: In this study, the segregation ratio of plants with non-pointed tip fruits and pointed tip fruits was 1,802:198 in the F_2 population. However, using the BSA analysis, we identified three loci underlying pointed tip fruit in the F_2 population. A previous study showed that all blossom-end morphology genes are inherited as recessives in tomato⁵. If these three loci did not interact with each other, the segregation ratio of non-pointed tip trait and pointed tip trait should be 63:1, which was far from our data. Thus, there could be specific genetic interactions among the three loci. To avoid misleading the readers, we deleted the "recessive nature of the pointed tip" in the revised manuscript.

- Line 159, replace speculated with hypothesized.

Re: It was corrected with "hypothesized".

- Line 180, the gene mutation is predicted to lead to a protein that has fewer C2H2 domains. Replace the word "alter" to "is predicted to alter"

Re: It was corrected with "is predicted to alter", accordingly.

- Line 185 and 187, reverse the statement changes an arginine to a histidine residue. Histidine is wild type and thus the mutation changes the histidine to an arginine.

Re: It was corrected with "histidine to an arginine", accordingly.

- Line 198, this paragraph is not starting appropriately. The first sentence should be part of the last paragraph ending on line 197. In addition, add that the promoters of the alleles of the parents in the biparental cross were sequenced and not the whole GWAS population.

Re: The sentence was corrected and moved to the last paragraph in the revised manuscript on lines 189 to 192.

- Line 204, remove "In fact, *PT* suppl figure S5A) because it is not necessary.

Re: We have added the *PT* transcript levels in more accessions at different stages spanning the pointed tip formation period, which would provide further evidence for the stage of fruit development at which the fruit pointed tips are initiated. Therefore, the sentence was revised, and it now reads "In fact, *PT* was most highly expressed in the leaf, flower, style, ovary, and early immature fruit (3-7 DPA) of both pointed tip and non-pointed tip fruit accessions (Supplementary Fig. 5)" in the revised manuscript.

- Remove the GUS data is figure S5B because the data are not convincing. Same for Fig 5C. The GUS expression pattern is weak and diffuse and not specific to the tip.

Re: The GUS data have been discarded in the revised manuscript.

- Line 208, "taken together..." is not correct. The expression of GUS or *PT* does not show a change in protein sequence. Please modify the sentence accordingly.

Re: We corrected the sentence to "Combining the analyses of variations and transcript levels, we concluded that the lead SNP in *PT* was predicted to alter the number of C2H2 domains in the *PT* protein and that this lead SNP was probably responsible for the formation of fruit with pointed tips" in the revised manuscript.

- Line 227, the shape of the fruits is more oval than pear. I would consider that overexpression of *PT^H* leads to pear or oval fruit depending on the genetic background. Or oval in both cases.

Re: We agree with the reviewer's comment that the shape of the fruits was more oval than pear. The sentence was corrected as suggested in the entire revised manuscript.

- Line 237, Specifically, *PT^R* promotes the formation of fruits with pointed tips is likely incorrect (see comments above). It is not likely based on the data shown that *PT^R* has a function in the plant.

Re: Based on the analysis of the phenotype of the F2:3 plants harboring either *PT^H* or *PT^R* and the CR-*pt^{H/R}* knockout lines, we confirmed that *PT^R* is a loss of function allele for fruit pointed tip in tomato. We have deleted the sentence "Specifically, *PT^R* promotes the formation of fruits with pointed tips" in the revised manuscript.

- Line 242, add the word endogenous before *PT*.

Re: It was added, accordingly.

- Line 247, "... that changed the reading frame.." is not correct. One mutant is a 9 bp deletion and that is an inframe mutation. The authors need to explain how an in frame mutation has the same affect as an out of frame mutation. Moreover, the authors should present in the supplement the resulting amino acid sequence in the CR lines for each allele. In one case, the gRNA seems almost in the intron, does it affect amino acid sequences or not?

Re: The predicted protein of the CR-*pt^H-4* line was a deletion of three amino acids in the conserved spacers between zinc fingers. This sentence was revised to "All selected lines contained either an insertion or a deletion in the target region of the *PT* gene that changed the reading frame or deletion of amino acid residues in the conserved spacer between zinc fingers and were thus, knockout mutants". We provided the predicted amino acid sequences of the CR lines in revised Supplementary Figs 8, 11, accordingly. Additionally, all the gRNAs used in this study were designed to correspond to sequences in the exons of *PT*.

- Line 254, the overexpression of *PT^H* leads to an oval shaped fruit. The term round edge at the distal end is ambiguous.

Re: The statement was corrected to "leads to the formation of a non-pointed tip fruit in tomato" in the revised manuscript.

- Fig 4E would benefit from using mutant probe sequence as well to determine if that can not compete with the wild type probe.

Re: Following your suggestion, we used a mutant probe to examine the binding ability of *PT^H* to the *FUL2* promoter in the EMSA assay. We found that the mutant probe could not bind to the *PT^H* protein. We added this data in revised Fig. 5e.

- Line 287, replace indicate with imply because the statement is too strong.

Re: It was corrected with "imply", accordingly.

- Line 315, rephrase the rounded edge terms as they are ambiguous.

Re: It was corrected with "non-pointed tip", accordingly.

- Line 318, add transport after auxin.

Re: We corrected it in the revised manuscript as suggested.

- The manuscript has several transitions that are odd. The transition from phenotype to *FUL2* and auxin is unclear. Yes, overexpression of *FUL2* leads to pointed fruit shape and yes auxin controls tons of processes

in plants including pointed fruit tips. Yet, focusing only on auxin transport (PINs and LAX) does not mean that auxin is controlling these processes exclusively because other processes were not ruled out. The unique role of auxin and *FUL2* would be more convincing if they were identified through an RNA seq experiments. The authors should show that GA or ethylene (for example) are not involved by qRT PCR or by RNA seq.

Re: Following the suggestion of this reviewer, we conducted an RNA-seq analysis on the pointed tip tissue of *CR-pt^H* and the corresponding pericarp tissue of the wild-type TS-9 control. We found that *FUL2* was significantly increased in the *CR-pt^H* lines as compared to its expression level in the wild-type TS-9. In addition, GO enrichment analysis of differentially expressed genes from the RNA-seq showed that a large portion of enriched terms were found to be associated with the auxin polar transport and auxin homeostasis. We provided this data in revised Supplementary Fig. 9.

- The role of auxin in regulating fruit shape could be strengthened by demonstrating that TS3 *pt^R* does not affect IAA levels.

Re: Following the advice of this reviewer, we quantified the endogenous auxin content in the pointed tips of the fruit from *CR-pt^R* lines and the wild-type TS-3. We found that the IAA, ICA, and ME-IAA contents of pointed tips from the fruits produced by the *CR-pt^R* lines were similar to their levels in the wild-type TS-3. We provided this data in revised Supplementary Fig. 13.

- Lines 338-347, since the expression of only some genes is evaluated, it is not possible to conclude that auxin transport is critical because the authors have not demonstrated that other processes and hormone genes are not altered.

Re: To address the criticism, we performed an additional RNA-seq analysis on the pointed tip tissue of *CR-pt^H* and the corresponding pericarp of the wild-type TS-9. GO enrichment analysis revealed that *PT* might regulate genes associated with the terms 'DNA-binding transcription factor activity', 'microtubule binding', and 'auxin efflux transmembrane transporter activity' that are involved in 'regulation of transcription', 'mitotic cell cycle phase transition', 'auxin efflux', 'auxin polar transport', 'auxin homeostasis' and response to 'light stimulus' (Supplementary Fig. 9). Therefore, we focused on the auxin transport and detected changes in *PIN* and *LAX* expression levels in the pointed tips.

- Line 348, Similarly the transition to ethylene is not strong. For example, why focus on *ACO1* and not other genes?

Re: To elucidate the possible mechanism used by *PT^R* to regulate fruit ripening, we measured the expression levels of most fruit ripening-related genes in the fruit from *PT^R*-overexpressing lines and wild-type TS-9. Among these genes, only the *ACO1* expression level was significantly upregulated in the *PT^R*-overexpressing lines relative to TS-9. Thus, we focused on the expression of *ACO1* gene in *PT^R*-mediated early fruit ripening. As explained above, we have deleted the data and description of the fruit ripening parts in the revised manuscript.

- Fig 6B should show ethylene accumulation at an earlier and same time point as in A, 24 dpa. The ethylene accumulation experiment should also be conducted in the NILs.

Re: Indeed, to confirm the function of the PT^R allele in early fruit ripening, we should investigate ethylene accumulation in NILs. However, the regulation of fruit ripening is beyond the scope of this manuscript and we have discarded the fruit ripening part from the revised manuscript, following Reviewers' suggestions.

- Explain in fig 3H the meaning of the black dotted line.

Re: The black dotted lines in Fig. 3f-h represent the base sequences between sgRNAs. We added this information in the legend of revised Fig. 3.

- Figure 3, For PT's role in ripening, the authors should present a controlled fruit ripening experiment. Namely, fruits need to be tagged and ripening in dpa should be recorded in all transgenic lines as well as the NILs.

Re: We have discarded the fruit ripening part from the revised manuscript (see above reply).

- The dpa should be shown in all figure panels that show images of fruit at different maturation stages.

Re: We have discarded the fruit ripening part from the revised manuscript (see above reply).

- The legends for most figures could be improved (Representative pieces of fruit, for example, is incorrect phrasing).

Re: "Representative pieces of fruit" were changed with "Phenotype of fruit". Additionally, we checked all the figure legends and revised them accordingly.

- Figure 5, the role of auxin inhibitors would be much better studied in a normal tomato such as the parents of the mapping study. Microtom has hormone mutations to accommodate its compact growth and these could interact with auxin in regulating the pointed phenotype. Does Microtom carry the PT^R allele? In any case, the authors should repeat the experiment with the lines used in the mapping population.

Re: Yes, Micro-Tom carries the PT^R allele. Following the suggestion of this reviewer, we have replaced it with the parents of the mapping population. We provided this data in revised Figs. 6a-e.

References

1. Lu J, *et al.* OBV (obscure vein), a C2H2 zinc finger transcription factor, positively regulates chloroplast development and bundle sheath extension formation in tomato (*Solanum lycopersicum*) leaf veins. *Hortic Res-England* **8**, 230 (2021).
2. Jones CM, Rick CM, Adams D, Jernstedt J, Chetelat RT. Genealogy and fine mapping of *obscuravenosa*, a gene affecting the distribution of chloroplasts in leaf veins, and evidence of selection during breeding of tomatoes (*Lycopersicon esculentum*; Solanaceae). *American journal of botany* **94**, 935-947 (2007).
3. Soyk S, *et al.* Variation in the flowering gene SELF PRUNING 5G promotes day-neutrality and early yield

- in tomato. *Nature genetics* **49**, 162-168 (2017).
4. Razifard H, *et al.* Genomic Evidence for Complex Domestication History of the Cultivated Tomato in Latin America. *Molecular biology and evolution* **37**, 1118-1132 (2020).
 5. Barten J, Scott J, Gardner R. Characterization of blossom-end morphology genes in tomato and their usefulness in breeding for smooth blossom-end scars. *Journal of the American Society for Horticultural Science* **119**, 798-803 (1994).
 6. Wang S, *et al.* Members of the tomato FRUITFULL MADS-box family regulate style abscission and fruit ripening. *Journal of Experimental Botany* **65**, 3005-3014 (2014).
 7. Chunli ZHANG TW, Jing LI, Danqiu ZHANG, Qingmin XIE, Shoaib MUNIR, Jie YE, Hanxia LI, Yongen LU, Changxian YANG, Bo OUYANG, Yuyang ZHANG, Junhong ZHANG, Zhibiao YE. FUNCTIONAL GAIN OF FRUIT NETTED-CRACKING IN AN INTROGRESSION LINE OF TOMATO WITH HIGHER EXPRESSION OF THE *FNC* GENE. *Front Agr Sci Eng* **8**, 280-291 (2021).
 8. Shang L, *et al.* A mutation in a C2H2-type zinc finger transcription factor contributed to the transition toward self-pollination in cultivated tomato. *The Plant cell* **33**, 3293-3308 (2021).
 9. Bemer M, *et al.* The tomato FRUITFULL homologs TDR4/FUL1 and MBP7/FUL2 regulate ethylene-independent aspects of fruit ripening. *The Plant cell* **24**, 4437-4451 (2012).
 10. Lin T, *et al.* Genomic analyses provide insights into the history of tomato breeding. *Nature genetics* **46**, 1220-1226 (2014).

Reviewers' Comments:

Reviewer #1:

Remarks to the Author:

The authors have revised thoroughly the manuscript and have added new data. They confirm and improve their molecular tests and the conclusions appear now solid. However, regarding the evolutionary analysis, the revision has been only cosmetic in the result section. The new information from the Lu et al. and other papers suggest that indeed selection has likely not been at the PT locus, but rather as a side effect of selection at other loci in this genomic region. The authors state this now correctly on lines 491-492. As seen in figure 7, a large cluster of genes likely hitch-hiked with SP5G. My main comment is thus to ask for careful phrasing of the sentence in the abstract (line 26-27) and of the result section related to the evolutionary analysis in order to remove any ambiguity. I suggest some possible example of rephrasing below for rigorous claims.

Abstract, Line 26-27: "Our evolutionary analysis combined with other studies indicated that the PT_R allele likely hitch-hiked along with other selected loci during the domestication process."

Results section, Lines 375-389

"To determine whether PT could have been a target of selection in the process of tomato domestication and improvement, we evaluated the nucleotide diversity (π) between groups that we previously evaluated. Comparing the ratios of nucleotide diversity between PIM and CER accessions, we found that PT was part of a large genomic cluster with high ratio of diversity, indicating one or several selection events during tomato domestication (Figs 7a, b). Additionally, as PT is part of a previously reported domestication sweep, our results suggest that the mutation/phenotype likely hitch-hiked along with other loci sweeping through. A particular candidate for selection is the SP5G locus. Meanwhile, we determined the PT H and PT R genotypes in 311 tomato accessions (Supplementary Data 1). The frequency of the PT H and PT R alleles in the accessions from the wild tomato group was 98.2% and 1.8%, respectively. The frequency of PT R gradually increased in the CER (8.6%) and BIG (24.5%) groups (Fig. 7c). We found that the PT R (G) allele was mainly distributed in the United States, Russia, and along the Mediterranean coast (e.g., in Italy), consistent with possible hitch-hiking with other loci under selection in this large genomic region during the process of tomato domestication and later on improvement (Fig. 7d).

Reviewer #2:

Remarks to the Author:

The revised manuscript re-submitted by Song et al. adequately addresses, in most cases, the critical points raised in the review of the first version. Overall, the manuscript has been substantially improved, although some important issues still remain to be solved.

1- It is important that the authors should clearly refer from the beginning (from the introduction) to the antecedent work published by another group (Lu et al. 2021) which describes the same mutation within the same locus and mention the previous cloning of the "Obscure Vein" gene which turns out to be also the "Pointed Tip" gene. The manuscript cannot ignore this fact and retain the same form of writing as when the authors were unaware of the existence of this work. I even suggest that it would be more appropriate to name the gene OBV/TP, as this would be more in line with reality and thus provide better clarity to readers. To my opinion, referring properly to the important work of Lu et al. 2021 doesn't diminish the novelty of the findings dealing with the pointed tip described in the manuscript. It rather provides an interesting new functional significance to the OBV/PT gene that has not been uncovered so far.

2- In connection with the previous comment, the authors should elaborate more, in the Results and Discussion sections, on the obscure vein phenotypes in their mutants and compare to those reported in Lu et al. 2021.

3- Another point is related to their statement that the pointed tip has been selected during domestication as a target locus to improve this trait. The evolutionary analysis doesn't seem to support unambiguously this hypothesis, so the authors should tone down their claim in this regard as their phrasing remains ambiguous. Most likely, the PT gene has been selected inadvertently by a hitch-hiking mechanism probably while targeting the SP locus.

Minor points:

- The discussion is too long and sometimes repeats what is already mentioned in the Introduction and Results section. For instance, the part from line 392 to line 397 can be deleted without any loss of meaning, and the Discussion may start at line 397 with "Generally, utilization of germplasm with pointed tip fruit ...".
- Line 413, it is not true that PTR is a loss-of-function allele as it only leads to partial loss-of-function.
- Line 482, recently should be recent.

Reviewer #3:

Remarks to the Author:

My previous concerns have been addressed in this revision. Well done!

Reviewer #4:

Remarks to the Author:

If I have carefully read the revised manuscript and find that it has satisfactorily addressed my major concerns about the original submission. I have only a few minor comments:

line 74: Although the provided reference supports the notion that the pointed tip could be a solution for the often large blossom end scar, I wouldn't call it "common" method just yet. Just "method" seems more appropriate

line 119-120: this line is confusing because the two alleles do not control a "transition" but a distinct phenotype: pointed or not, and the order of the two alleles does not reflect the phenotype pointed and non-pointed. The next line says it clearly, so in this line I suggest to say: control fruit blossom end shape

line 289: "the promoter of FUL2": clearly state what sequence length relative to the TSS or to the ATG (in that case not just promoter) has been used

line 305-307: also here clearly state the relative position of the fragment used for EMSA

line 481: "could be a potential target" suggest to replace by "could have been a target"

line 482: replace "a recently research" by "recent research"

line 483: insert "as" underlying the etc.

Line 491: "that PT, namely OBV," is a strange way to describe synonyms. "PT/OBV" would be better

line 493 to 498 is confusing. In the original reference 13 and in your introduction it is stated that a pointed tip can mitigate large blossom end scars and therefore is interesting to breeders. Tips or beaks may lead to damage during transport, if that is what you mean, but in itself, of course, does not increase transportation costs.

Reviewer #5:

Remarks to the Author:

With the addition of the RNA seq and its analysis, I have no further comments for the authors.

Point-by-point responses to Reviewers' comments

Reviewer #1 (Remarks to the Author):

The authors have revised thoroughly the manuscript and have added new data. They confirm and improve their molecular tests and the conclusions appear now solid. However, regarding the evolutionary analysis, the revision has been only cosmetic in the result section. The new information from the Lu et al. and other papers suggest that indeed selection has likely not been at the *PT* locus, but rather as a side effect of selection at other loci in this genomic region. The authors state this now correctly on lines 491-492. As seen in figure 7, a large cluster of genes likely hitch-hiked with *SP5G*. My main comment is thus to ask for careful phrasing of the sentence in the abstract (line 26-27) and of the result section related to the evolutionary analysis in order to remove any ambiguity.

I suggest some possible example of rephrasing below for rigorous claims.

Abstract, Line 26-27: "Our evolutionary analysis combined with other studies indicated that the *PT^R* allele likely hitch-hiked along with other selected loci during the domestication process."

Results section, Lines 375-389

"To determine whether *PT* could have been a target of selection in the process of tomato domestication and improvement, we evaluated the nucleotide diversity (π) between groups that we previously evaluated. Comparing the ratios of nucleotide diversity between PIM and CER accessions, we found that *PT* was part of a large genomic cluster with high ratio of diversity, indicating one or several selection events during tomato domestication (Figs 7a, b). Additionally, as *PT* is part of a previously reported domestication sweep, our results suggest that the mutation/phenotype likely hitch-hiked along with other loci sweeping through. A particular candidate for selection is the *SP5G* locus. Meanwhile, we determined the *PT^H* and *PT^R* genotypes in 311 tomato accessions (Supplementary Data 1). The frequency of the *PT^H* and *PT^R* alleles in the accessions from the wild tomato group was 98.2% and 1.8%, respectively. The frequency of *PT^R* gradually increased in the CER (8.6%) and BIG (24.5%) groups (Fig. 7c). We found that the *PT^R* (G) allele was mainly distributed in the United States, Russia, and along the Mediterranean coast (e.g., in Italy), consistent with possible hitch-hiking with other loci under selection in this large genomic region during the process of tomato domestication and later on improvement (Fig. 7d).

Reply: We apologize for the ambiguous description. Based on the reviewer's suggestions, we rephrased the sentences in the revised Abstract on lines 27 to 28 and the Results section on lines 378 to 394.

Reviewer #2 (Remarks to the Author):

The revised manuscript re-submitted by Song et al. adequately addresses, in most cases, the critical points raised in the review of the first version. Overall, the manuscript has been substantially improved, although some important issues still remain to be solved.

1- It is important that the authors should clearly refer from the beginning (from the introduction) to the antecedent work published by another group (Lu et al. 2021) which describes the same mutation within the same locus and mention the previous cloning of the "*Obscure Vein*" gene which turns out to be also the

"Pointed Tip" gene. The manuscript cannot ignore this fact and retain the same form of writing as when the authors were unaware of the existence of this work. I even suggest that it would be more appropriate to name the gene *OBV/TP*, as this would be more in line with reality and thus provide better clarity to readers. To my opinion, referring properly to the important work of Lu et al. 2021 doesn't diminish the novelty of the findings dealing with the pointed tip described in the manuscript. It rather provides an interesting new functional significance to the *OBV/PT* gene that has not been uncovered so far.

Re: In this revision, we added the description "This locus has also been identified recently to underlie the 'obscure vein' (*OBV*)." in the revised Introduction section on lines 118 to 119. Additionally, we mentioned the gene name "*PT/OBV*" in the revised Discussion section on line 491.

2- In connection with the previous comment, the authors should elaborate more, in the Results and Discussion sections, on the obscure vein phenotypes in their mutants and compare to those reported in Lu et al. 2021.

Re: We appreciate the suggestion. Even though *PT* and *OBV* are the same gene, the regulation of leaf vein development by *OBV* depends on a specific background, such as Micro-Tom. In our study, most of the accessions harboring the *PT^H* allele in the population did not produce transparent veins, which is inconsistent with those reported in Lu et al. 2021. However, consistent with their observations, we also detected the phenotype of transparent veins in *PT^H*-overexpressing plants. Therefore, we just elaborated the transparent veins phenotypes of *PT^H*-overexpressing plants in the Discussion section on lines 482 to 492. This is also the reason why we did not completely change the gene name of *PT* to *PT/OBV* in the full text.

3- Another point is related to their statement that the pointed tip has been selected during domestication as a target locus to improve this trait. The evolutionary analysis doesn't seem to support unambiguously this hypothesis, so the authors should tone down their claim in this regard as their phrasing remains ambiguous. Most likely, the *PT* gene has been selected inadvertently by a hitch-hiking mechanism probably while targeting the *SP* locus.

Re: We are sorry for the ambiguous description. Indeed, *PT/OBV* probably has "hitchhiked" along with selection for mutations in the nearby *SP5G*. Based on the reviewers' suggestions, we toned down the claim and rephrased them in the revised Abstract on lines 27 to 28 and the Results section on lines 378 to 394.

Minor points:

- The discussion is too long and sometimes repeats what is already mentioned in the Introduction and Results section. For instance, the part from line 392 to line 397 can be deleted without any loss of meaning, and the Discussion may start at line 397 with "Generally, utilization of germplasms with pointed tip fruit ...".

Re: As suggested, we have deleted this part in the revised Discussion section of the manuscript.

- Line 413, it is not true that *PT^R* is a loss-of-function allele as it only leads to partial loss-of-function.

Re: We have corrected it and the statement now reads "*PT^R* is a partial loss-of-function allele" in the

revised manuscript on line 412.

- Line 482, recently should be recent.

Re: It was corrected with "recent".

Reviewer #3 (Remarks to the Author):

My previous concerns have been addressed in this revision. Well done!

Re: We thank the reviewer again for reviewing this manuscript.

Reviewer #4 (Remarks to the Author):

If have carefully read the revised manuscript and find that it has satisfactorily addressed my major concerns about the original submission. I have only a few minor comments:

line 74: Although the provided reference supports the notion that the pointed tip could be a solution for the often large blossom end scar, I wouldn't call it "common" method just yet. Just "method" seems more appropriate

Re: It is corrected as " a method", accordingly (line 76).

line 119-120: this line is confusing because the two alleles do not control a "transition" but a distinct phenotype: pointed or not, and the order of the two alleles does not reflect the phenotype pointed and non-pointed. The next line says it clearly, so in this line I suggest to say: control fruit blossom end shape

Re: Thank you for pointing this out. Based on your suggestion, we corrected it with "control the fruit blossom end shape" in the revised manuscript on line 120.

line 289: "the promoter of FUL2": clearly state what sequence length relative to the TSS or to the ATG (in that case not just promoter) has been used

Re: The promoter of "1, 504 bp upstream of the transcription start site" was used for yeast one-hybrid assay. We added this description in the revised manuscript on line 290.

line 305-307: also here clearly state the relative position of the fragment used for EMSA

Re: The fragment (-1,115 bp to -1,066 bp) was labeled and used for EMSA. This description is now added as suggested (line 309).

line 481: "could be a potential target" suggest to replace by "could have been a target"

Re: It is corrected with "could have been a target", accordingly (line 378).

line 482: replace "a recently research" by "recent research"

Re: It is corrected with "recent research", as suggested (line 482).

line 483: insert "as" underlying the etc.

Re: It is added, accordingly (line 483).

Line 491: "that PT, namely OBV," is a strange way to describe synonyms. "PT/OBV" would be better

Re: It is corrected as suggested (line 491).

line 493 to 498 is confusing. In the original reference 13 and in your introduction it is stated that a pointed tip can mitigate large blossom end scars and therefore is interesting to breeders. Tips or beaks may lead to damage during transport, if that is what you mean, but in itself, of course, does not increase transportation costs.

Re: We apologize for the confusing description. We mean that people will pay more to prevent fruit from damaging during the processes of storage and transportation. To avoid misunderstanding, we deleted the statement "Fruit with the pointed tip will increase the storage and transportation costs, and rough blossom-end scarring severely reduces the quality of tomatoes." in the revised manuscript (line 494).

Reviewer #5 (Remarks to the Author):

With the addition of the RNA seq and its analysis, I have no further comments for the authors.

Re: We thank the reviewer again for reviewing this manuscript.